



**Bedrock depth influences spatial patterns of summer baseflow, temperature, and flow**
**disconnection for mountainous headwater streams**
Martin A. Briggs[1*]
Phillip Goodling[2]
Zachary C. Johnson[3]
Karli M. Rogers[4]
Nathaniel P. Hitt[4]
Jennifer B. Fair[4,5]
Craig D. Snyder[4]
[1]U.S. Geological Survey, Earth System Processes Division, Hydrogeophysics Branch, 11
Sherman Place, Unit 5015, Storrs, CT 06269 USA
[2]U.S. Geological Survey, Maryland-Delaware-District of Colombia Water Science Center, 5522
Research Park Drive, Catonsville, MD, 21228, USA
[3]U.S. Geological Survey, Washington Water Science Center, 934 Broadway, Suite 300, Tacoma,
WA 98402 USA
[4]U.S. Geological Survey, Eastern Ecological Science Center, 11649 Leetown Road,
Kearneysville, WV 25430 USA
[5]U.S. Geological Survey, New England Water Science Center, 10 Bearfoot Road, Northborough,
MA 01532 USA
Corresponding author: Martin Briggs, mbriggs@usgs.gov



**Abstract**
In mountain headwater streams the quality and resilience of cold-water habitat is regulated by
surface stream channel connectivity and groundwater exchange. These critical hydrologic
processes are thought to be influenced by the stream corridor bedrock contact depth (sediment
thickness), which is often inferred from sparse hillslope borehole information, piezometer
refusal, and remotely sensed data. To investigate how local bedrock depth might control summer
stream temperature and channel disconnection (dewatering) patterns, we measured stream
corridor bedrock depth by collecting and interpreting 191 passive seismic datasets along eight
headwater streams in Shenandoah National Park (Virginia USA). In addition, we used multiyear
stream temperature and streamflow records to calculate summer baseflow metrics along and
among the study streams. Finally, comprehensive visual surveys of stream channel dewatering
were conducted in 2016, 2019, and 2021 during summer baseflow conditions (124 total km of
stream length). We found that measured bedrock depths were not well-characterized by soils
maps or an existing global-scale geologic dataset, where the latter overpredicted measured
depths by 12.2 m (mean), or approximately four times the average bedrock depth of 2.9 m. Half
of the eight study stream corridors had an average bedrock depth of less than 2 m. Of the eight
study streams, Staunton River had the deepest average bedrock depth (3.4 m), the coldest
summer temperature profiles, and substantially higher summer baseflow indices compared to the
other study steams. Staunton River also exhibited paired air and water annual temperature signals
suggesting deeper groundwater influence, and the stream channel did not dewater in lower
sections during any baseflow survey. In contrast, streams Paine Run and Piney River did show
pronounced, patchy channel dewatering, with Paine Run having dozens of discrete dry channel
sections ranging 1 to greater than 300 m in length. Stream dewatering patterns were apparently



influenced by a combination of discrete deep bedrock (20 m+) features and more subtle sediment
thickness variation (1-4 m), depending on local stream valley hydrogeology. In combination
these unique datasets show the first large-scale empirical support for existing conceptual models
of headwater stream disconnection based on underflow capacity and shallow groundwater
supply.



## 1. Introduction

Mountain headwater stream habitat is influenced by hydrologic connectivity along the surface channel, and connectivity between the channel and multiscale groundwater flowpaths (Covino, 2017; Wohl, 2017). Discharge from shallow groundwater within the critical zone is a primary component of stream baseflow, attenuating maximum summer temperatures and creating cold water habitat (Singha and Navarre-Sitchler, 2021; Sullivan et al., 2021). In headwater stream valleys characterized by irregular bedrock topography and thin, permeable sediments, nested physical processes interact to control the connectivity of groundwater/surface water exchange (Tonina and Buffington, 2009). Between stormflow and snowmelt events, headwater streamflow (baseflow) is primarily generated by groundwater discharge due to a relative lack of soil water storage and release (Winter et al., 1998). Unlike in lower valley settings, mountain headwaters accumulate reduced fine soil, facilitating efficient routing of quickflow to streams through macropores and other preferential flowpaths within regolith and saprolite (Sidle et al., 2000). Recharge that does percolate vertically contributes to shallow groundwater along steep hillslopes and valley floors, where groundwater flowpath depths are constrained by bedrock topography (Buttle et al., 2004). Although deeper groundwater may also represent an important contribution to summer streamflow in systems with relatively permeable bedrock (Burns et al., 1998; O'Sullivan et al., 2020), shallow, low permeability bedrock generally restricts stream-groundwater connectivity to the thin layers of unconsolidated sediments (Briggs et al., 2018b).

In addition to baseflow drainage along headwater stream networks, down-valley shallow groundwater 'underflow' can be substantial when high gradient streams lack sinuosity and flow over permeable sediment (Figure 1a, Figure A1). In fact, headwater stream channels may only be expected to show surface flow when the transmission ability of the underlying alluvium and



colluvium is exceeded, and bedrock depth is thought a primary control of this underflow capacity
(Ward et al., 2020). In some hydrogeologic settings, underflow can dominate groundwater export
from mountain catchments compared to groundwater drainage via the surficial stream channel
(Larkin and Sharp, 1992; Tiwari et al., 2017). Moreover, in addition to longitudinal transport
down-valley, underflow also acts as a reservoir of exchange for hyporheic flowpaths that may
mix with shallow groundwater before returning to channel flow (Payn et al., 2009), transporting
buffered temperature signals back to channel waters (Wu et al., 2020). Local underflow is
recharged from upgradient flowpaths and adjacent hillslopes, creating complex seasonal and
interannual patterns in groundwater connectivity and discharge to surface water (Jencso et al.,
2010; Johnson et al., 2017). A major challenge to understanding groundwater exchange in
headwaters is that attributes of the streambed subsurface, such as the depth to the underlying
bedrock contact, are often only available from limited direct measurements, coarse spatial
interpolations, or inferred remotely based on landscape forms. Therefore, methods that allow
efficient, local measurements of the streambed subsurface are critically needed.

Seasonal thermal regimes of mountain headwater streams can be profoundly impacted by

groundwater inflow from multiple depths (Briggs et al 2018a). In lower valley settings, the
temperature of groundwater discharge along stream networks is often assumed to approximate
the average annual land surface temperature throughout the year (Stonestrom and Constantz,
2003). Conversely, shallow groundwater temperature (within several m from land surface) can
show pronounced seasonality (Bundschuh, 1993; Lapham, 1989) and high spatial variability,
even over small spatial extents (Snyder et al. 2015). The warming of shallow groundwater during
the summer and fall seasons can limit the ability of gaining mountain streams to support cold-
water fish populations during the low flow season, even if baseflow (assumed to be dominated





by groundwater discharge) fractions are large (Johnson et al., 2020). In systems with low
permeability bedrock, thicker hillslope sediments may generate deeper, colder lateral
groundwater flow to streams in summer (Figure 1a), increasing cold water habitat resiliency
(Briggs et al., 2018b). For example, a recent meta-analysis of stream and air temperature records
across the contiguous United States found that a substantial fraction of shallow groundwater
dominated streams displayed summer warming trends in recent decades, while deeper
groundwater dominated streams were more stable (Hare et al., 2021). Steep mountain stream
systems such as those found in the Blue Ridge and Cascade mountains of the USA have been
found to show annual thermal regimes dominated by the annual thermal signals of shallow
groundwater (Johnson et al., 2020), indicating such streams may also be at risk for warming over
time, contrary to assumptions based on elevation alone.

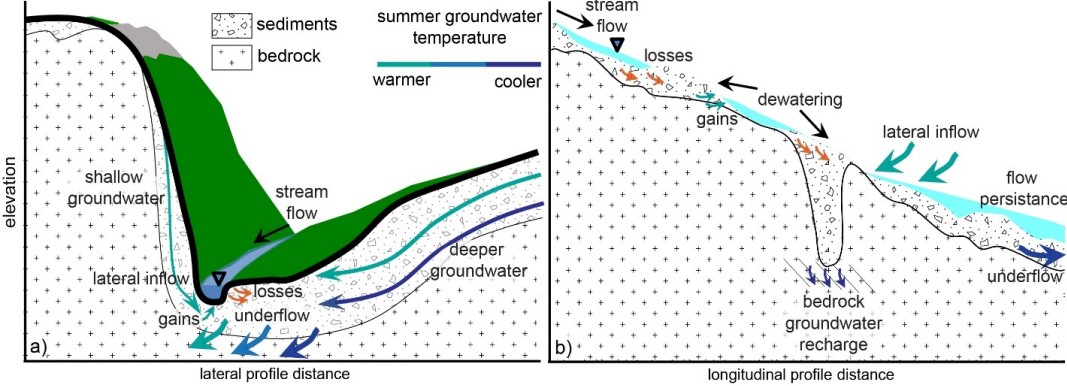

*Figure 1. A conceptual mountain stream valley cross section (panel a) and longitudinal profile*
*(panel b) indicating the expected control of low permeability bedrock topography on*
*groundwater temperature, stream-groundwater exchange, patchy stream dewatering, and the*
*underflow reservoir.*
Beyond warm summer stream temperatures, the dewatering and disconnection of the

active stream channel during summer low flows can adversely impact fish habitat by impeding
fish movement (Edge et al., 2017; Labbe & Fausch, 2000;  Rolls et al., 2012; Snyder et al.,
2013), locally degrading water quality (Hopper et al., 2020), and increasing predation risks in


isolated pools (Magoulick and Kobza 2003). However, the physical controls on localized stream
channel dewatering are not well characterized and likely involve a spectrum of nested gaining
and losing flowpaths. For mountain headwater streams, previous research has documented
general seasonal shifts in hydraulic gradients from gaining to losing, with closely coupled
streamflow and precipitation events, indicating a dominance of shallow routing rather than
deeper groundwater connectivity in maintaining streamflow (Zimmer and McGlynn, 2017).
Locally-losing sections of headwater stream channels can be associated with coarse, permeable
colluvial deposits from hillslope mass wasting processes (Costigan et al., 2016; Weekes et al.,
2015), as local enhancement of the total pore space under mountain streams can drive
downwelling of streamwater (Figure 1b, Tonina & Buffington, 2009). Main channel dewatering
occurs when the bed sediments have a storage and transport capacity that exceeds stream
discharge (Rolls et al., 2012; Ward et al., 2018), though stream water losses can also be driven
by local changes in bed morphology and slope (Costigan et al., 2016). The shallowing of the
underlying bedrock contact may drive lateral underflow toward the surface causing the channel
to gain water (Herzog et al., 2019)(Figure 1b), though such hypothesized dynamics are not well
documented in existing literature due to a relative lack of bedrock topography data along
headwater streams.

At large scales, contiguous bedrock depth layers are interpolated from a combination of

relatively sparse borehole data and surface topography (Kauffman et al., 2018; Pelletier et al.,
2016; Shangguan et al., 2017). However, in steep headwater systems with little borehole data,
bedrock topography is difficult to predict accurately from land surface topography alone. The
development of improved tools for predicting bedrock depth is an active area of research which
has recently demonstrated promise when bedrock outcrop data are included (e.g. Furze et al.,



2021; Odom et al., 2021). The limitations of using landform data to predict bedrock depth are
compounded by inherent challenges in collecting physical data via soil pits and monitoring wells
in rugged, rocky terrain, and so direct measurement data are often limited to highly studied
experimental watersheds where bedrock depth is inferred from piezometer installation refusal
(e.g. Jencso et al., 2010; Ward et al., 2018).

Application of near surface geophysical methods to stream corridor research has

increased appreciably in recent years (McLachlan et al., 2017), and several methods are sensitive
to shallow subsurface flow and geologic attributes including bedrock depth. Active seismic
refraction measurements can provide high resolution (10s of cm) bedrock depth information
along transect-based cross-sections (e.g. Flinchum et al., 2018), but are less suited for
exploration throughout rugged mountain stream valleys at the many km-scale due to logistical
challenges in using active seismic methods to obtain a sufficient amount of data to effectively
characterize important variation in bedrock depth at relatively small, ecologically-relevant spatial
scales. Point-based, efficient passive seismic measurements represent a unique combination of
high mobility and relative precision for measuring bedrock depth along mountain valleys. The
horizontal-to-vertical spectral ratio (HVSR) method is a passive seismic technique that evaluates
ambient seismic noise recorded using handheld instruments placed on the ground surface to
identify seismic resonance, which occurs at distinct unconsolidated sediment/bedrock interfaces
(Yanamaka et al., 1994).

The control of stream to groundwater exchange (i.e. 'transmission losses') on streamflow

permeance has been highlighted as an important research need by the comprehensive review of
intermittent stream systems by Costigan et al., (2016). Following the conceptual model of Ward
et al., (2018), a central hypothesis of our research was that bedrock depth along the stream



corridor will act as a first-order control on stream dewatering patterns when shallow bedrock is
of low permeability. Based on the concepts presented by Tonina & Buffington, (2009), we
postulated that relatively thick, permeable surficial sediment zones could locally accommodate
the entirety of low streamflow volumes, dewatering main channel sections at varied scales when
not balanced by groundwater inflow (Figure 1b). We further hypothesized that summer stream
channel thermal regimes would also be influenced by bedrock depth, as the temperature of
groundwater flowpaths that generate baseflow is depth dependent (Briggs et al., 2018b),
indicated conceptually in Figure 1a. To test our hypotheses, we extended the existing mountain
headwater bedrock depth surveys from Shenandoah National Park (SNP), Virginia, USA to
seven additional subwatersheds and compared results to physical mapping of stream dewatering,
multi-year stream temperature data and derived groundwater influence metrics, and baseflow
separation analysis to address the following research questions:
1. Does stream corridor bedrock depth exhibit longitudinal spatial structure in mountainous
streams? Can measured bedrock depth dynamics be accurately extracted from existing large-
scale datasets or inferred from high resolution soils maps?
2. Does underflow generally represent a net source or sink of summer flow for headwater
streams based on observed dewatering patterns and groundwater influence metrics?
3. Does bedrock depth explain spatial variation in stream temperature and summer baseflow
indices within headwater streams?
**2. Study Area**

The SNP is an 800 km$^2$ area of preserved headwater forest perched along a major

ridgeline of the Blue Ridge Mountains in northern VA, USA (Figure 2). The bedrock of the park
is predominantly low permeability basaltic and granitic material in the central and northern



sections, and siliciclastic along the southern section (Southworth et al., 2009), though many
subwatersheds also transition in dominant bedrock type. Stream valleys of SNP are typically
steep and feature a perennial channel with mainly non-perennial tributaries (Johnson et al.,
2017), Figure A1) and stream baseflow consists of less than 3-yr old groundwater on average
(Plummer et al., 2001). In contrast, water collected from SNP hillslope wells completed in
shallow fractured rock generally have higher ages of 10-20 yr (Plummer et al., 2001), indicating
minimal contributions from bedrock groundwater to streamflow. Previous ecohydrological
research in SNP has noted that some mainstem stream channels show patchy dewatering at
summer low flows (Snyder et al., 2013), though the physical controls on these patterns of stream
drying were not clear.

In SNP, stream baseflow is thought to be predominantly generated by near-surface

drainage of coarse unconsolidated alluvium and colluvium (DeKay, 1972; Nelms and Moberg,
2010). The mountain ridgeline streamflow systems are expected to drain near-surface flowpaths
and accommodate substantial down valley underflow below perennial stream channels (Figure
A1). A portion of hillslope recharge is expected to percolate downward through connected
bedrock fractures into the deeper groundwater reservoir contributing to mountain block recharge
along the Shenandoah River Valley. Narrow alluvium deposits mapped along the stream
corridors of SNP are thought to generally range up to 6 m in thickness and be more clay rich
when sourced by basaltic bedrock (Southworth et al., 2009). Data at sparse wells drilled along
the SNP ridgeline indicate bedrock depth can range over 20 m on hillslopes and be highly
variable (DeKay, 1972; Goodling et al., 2020; Lynch, 1987).

Previous research has inferred summer and annual groundwater discharge patterns

throughout SNP subwatersheds based on paired, local air and stream water temperature



dynamics (Briggs et al., 2018a; Johnson et al., 2017; Snyder et al., 2015). Combined, these
analyses indicated groundwater exchange is highly variable in space along singular stream
valleys and between subwatersheds, and dependent upon local- to subwatershed-scale
characteristics. A combination of landform features that include stream slope and stream valley
confinement operate in conjunction with seasonal precipitation to drive groundwater influence
on summer stream temperatures (Johnson et al., 2017). Multi-week lags in time between
streamwater and local air annual temperature signals (i.e. phase shifts toward later time) were
observed from dozens of the 120 total monitored stream sites indicating a dominance of shallow
groundwater discharge, originating generally within approximately 3 m of land surface (Briggs
et al., 2018a).




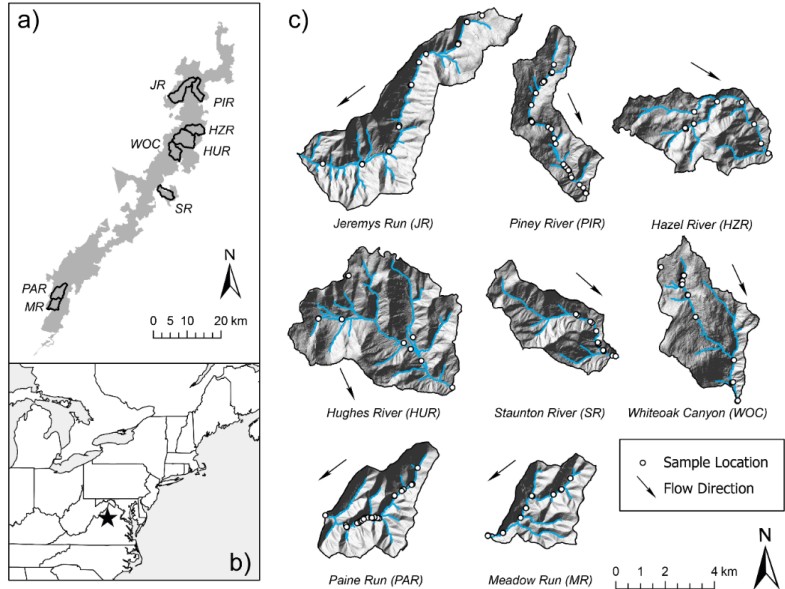

*Figure 2. This study was based in Shenandoah National Park (panel a) located in the Blue Ridge*
*Mountains of northeast USA (panel b). LiDAR hillshade cutouts of each subwatershed illustrate*
*the rugged terrain and varied valley morphology (panel c). The mainstem stream channel and*
*tributaries are traced and HVSR measurement locations noted.*
**3.0 Methods**
*3.1 Passive Seismic Bedrock Depth Measurements*
Periodically from the summer of 2016 to the spring of 2020, we acquired 323 HVSR
measurements across SNP. The geophysical data were collected along the perennial streams of
seven subwatersheds with extensive existing stream temperature and ecological datasets, and at
known ridgeline and hillslope borehole locations. This effort added to previously interpreted
HVSR data from 22 riparian sites collected along the Whiteoak Canyon subwatershed in late
2015 (Briggs et al., 2017), for a total of 8 mountain streams for analysis in this study (Figure 2).
In July 2016, HVSR data were collected in the following subwatersheds: Piney River, Paine
Run, Meadow Run, Jeremy's Run, Hazel River, and Hughes River.

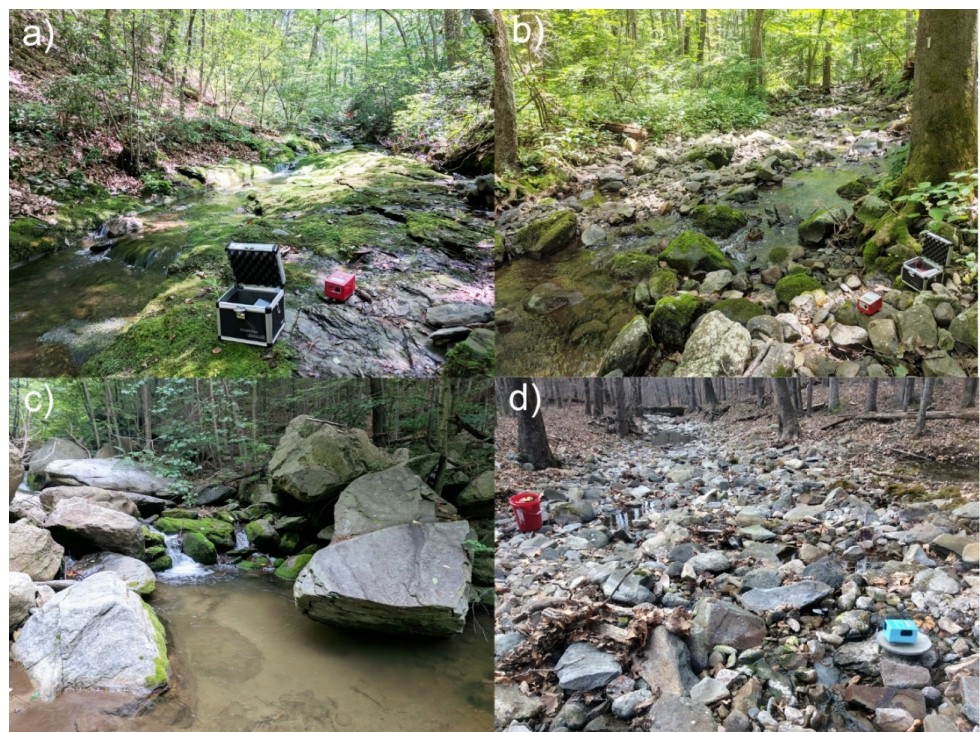

*Figure 3. Typical sections of a) Paine Run, b) Piney River, c) Staunton River, and d) a section of Paine Run that was dewatered at baseflow leaving isolated pools. The passive seismic HVSR instruments are shown deployed in panels a), b) and d) (Photographs by the U.S. Geological Survey).*

Measurement locations mostly coincided with existing stream temperature monitoring

stations (described by Snyder et al., 2017), and were typically made at points immediately

adjacent to the stream or on larger rocks within the channel (Figure 3). In July 2019, HVSR data

were again collected along Paine Run and Piney River subwatersheds, and throughout the lower

Staunton River (Figure 3). The 2019 survey design differed in that transect measurements were

made at 4 locations along the stream channel waterline spaced approximately 25 m apart at

longitudinal locations that differed from the 2016 survey. This was done to assess potential

variation in bedrock depth along short subreaches of these three streams. Finally, clustered

HVSR data were collected in March 2020 in Paine Run and Piney River in zones previously





258 observed to show channel disconnection and streamflow re-emergence. Measurement locations

259 were chosen to test the hypothesis that the dewatering patterns were controlled by bedrock depth

260 as shown conceptually in Figure 1b.

261  HVSR data were collected using multi-component Tromino seismometers (MOHO,

262 S.R.L.) directly coupled to the land surface or placed on heavy metal plates where sediment was

263 loose. Collection times ranged 10-20 min at either 128 or 256 Hz sampling rates. HVSR data

264 collection locations were determined by a combination of internal Tromino GPS and external

265 GPS units.  HVSR measurements were processed to derive a resonant frequency using a

266 commercially available program (GRILLA® v. 8.0 (2018); further details regarding data

267 processing are given by Goodling et al., (2020).

268  Resonant frequency measurements that passed a series of quality criteria were then

269 converted to a bedrock depth estimate following Briggs et al., (2017). This conversion

270 necessitates a shear wave velocity estimate for the unconsolidated sediments over bedrcok.

271 HVSR data collected at spatially distributed boreholes with documented depth to varied-type

272 bedrock along the SNP ridgeline indicated a shear wave velocity of 358.7 +/- 56 m/s (Goodling

273 et al., 2020). A similar shear wave velocity of 346 m/s was measured at two locations along the

274 Whiteoak Canyon riparian zone spaced several km apart using active seismic methods (Briggs et

275 al., 2018b). This agreement indicates a common shear wave velocity can be assumed for the

276 unconsolidated material of SNP subwatersheds. For this study we used the average of these

277 spatially distributed active and passive seismic methods at 352 m/s. The mean shear wave

278 velocity calculated in this study is comparable to the mean shear wave velocity ranges in firm

279 soils (180 - 360 m/s) and very dense soil and soft rock (360-760 m/s), according to National

280 Earthquake Hazards Reduction Program (NEHRP) guidelines (Building Seismic Safety Council,





1994). As an example of measurement sensitivity to the shear wave velocity parameter for
shallow bedrock contacts, a velocity change in either direction by 25 m/s would generally shift
the bedrock depth estimate by <0.2 m.
*3.2 Observations of spatial dewatering patterns*
Longitudinal (upstream to downstream) patterns of dewatering were determined in the

summers of 2016, 2019, and 2021 during baseflow conditions over 124 total km of stream length
for all surveys combined. In July-August of 2016 all eight subwatersheds (Figure 2) were
surveyed. In September of 2019 and August 2021, dewatering surveys were repeated in three
subwatersheds (Paine Run, Piney River, and Staunton River) to evaluate annual variation in
dewatering patterns. Data were collected by team of investigators walking each stream from an
upstream location defined by the point along the stream draining 75-hectares (assumed capture
area required to generate perennial streamflow, determined using watershed tools in ArcGIS) to
the bottom of each watershed near the park boundary, and mapping transition points between
three hydrologic categories: Wet, dry, or isolated pools based upon investigator observation.
"Wet" segments were defined as reaches where entire channel was wet with flow between pools;
"Dry" segments were defined as reaches containing no water, or isolated pools of insufficient
depth to sustain 1+ year old brook trout; and "Isolated Pools" were defined as reaches containing
pools of sufficient depth to support brook trout but were hydrologically disconnected from other
parts of the channel. An example of isolated pools is photographically depicted in Figure 3d.
Spatial coordinates of transition points were mapped using a Trimble R2 GNSS receiver for <1-
meter accuracy. Surveys for each subwatershed were completed within a single day to minimize
effects of temporal variation in precipitation.





In addition to local variability in bedrock depth, spatial patterns of dewatering and stream
temperature are likely to be influenced by seasonal precipitation and air temperature proximate
to the period of measurement (i.e., summer conditions, 2016 and 2019). We used historical
weather records (1942 – 2020) collected from the nearby Luray Weather Station located within
SNP (Station No. GHCND:USC00445096) to compare weather conditions during the two study
years with historical norms. Finally, 3D surface area of each subwatershed was determined from
existing LiDAR data using Add Surface Information in 3D Analyst Tools in ArcGIS and mean
valley width was evaluated from LiDAR data using 100-m transects measured approximately 2
m above the valley floor.
*3.3 Stream channel temperature data and baseflow separation*
Multi-year SNP stream temperature data were collected at hourly time intervals as
described by Snyder et al., (2017) using HOBO Pro V2 thermographs (+/- 0.2 °C expected
accuracy). From this larger dataset, 64 main channel locations within the 8 study subwatersheds
were extracted and processed for summary statistics such as the maximum and minimum of the
7-day running mean using Matlab R2019b software (Mathworks, Inc.). Only complete 7-day
periods were included in the running average. Warm season data (July, August, September) were
isolated and analyzed to coincide with the stream dewatering surveys and a larger body of
research regarding summer cold-water brook trout habitat in SNP. We utilized stream
temperature data processed by Briggs et al., (2018a) where dry sensor periods were identified
and removed, impacting a handful of the upper stream sites. Data were visualized and
downstream trends explored using Sigmaplot 14.0 software (Systat Software Inc.). Baseflow
separation was conducted for the three continuously gaged streams of this study (Paine Run,
Piney River, Staunton River) over summer months for the period of record (1993-2020).
Following the approach of (Hare et al., 2021), the daily Baseflow Index (BFI) was calculated





using the USGS-R 'DVstats' package (version 0.3.4) by dividing the calculated baseflow
discharge by the corresponding stream discharge, where a value of one would indicate stream
discharge was entirely composed of baseflow. BFI was then averaged (mean) across each
summer season, along with the mean and standard deviation of summer stream discharge.
**4. Results**
*4.1 Stream Corridor Bedrock depth*

Approximately 60% of individual HVSR measurements (191 of the 323) were of high

enough quality to be interpreted for bedrock depth using objective data quality metrics reported
by the GRILLA software. This ratio of interpretable to total HVSR measurements was similar to
the previous 2015 Whiteoak Canyon Run study using the same instrument type (Briggs et al.,
2017). For the 132 datasets that could not be interpreted, the primary reason was no identifiably
resonant frequency 'peak' in the multicomponent seismic data, as described in more detail in the
data release of Goodling et al., (2020). The loosely consolidated, rocky surficial soils of many
SNP subwatershed riparian zones likely contributed to poor instrument coupling to the land
surface, and therefore reduced measurement sensitivity/success compared to firmer soils.
However, due to spatial redundancy in the measurements, the 191 locations where bedrock depth
was evaluated generally covered all the intended longitudinal stream measurement locations
throughout the subwatersheds.




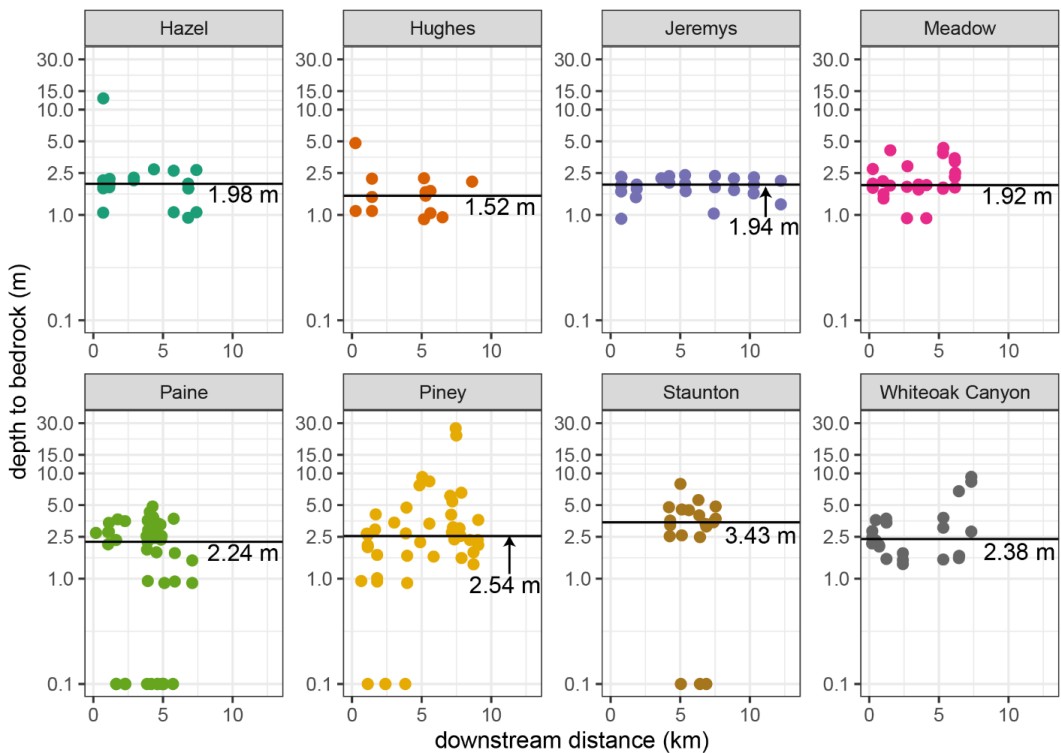

Figure 4. *Measured depth to rock along the stream channel and riparian zones of the eight study*
*subwatersheds. Exposed bedrock (i.e., zero depth) observed at the intended measurement*
*location is noted here by a value of '0.1' on the log scale. The median value is shown as a*
*labelled horizontal line.*

The median bedrock depth was smallest for Hughes River (1.52 m), and similar for

Meadow Run, Jeremy's Run, Hazel River (1.92, 1.94, 1.98, respectively, Table 1, Table B1,

Figure 4). Paine Run had a median of 2.24 m, Whiteoak Canyon of 2.38 m, and Piney River of

2.54 m. Lower Staunton River had the largest median depth to rock of 3.43 m (Table 1). Piney

River had the largest variation in bedrock depth, including a discrete zone greater than 20 m

deep, along with several zones of exposed bedrock along the channel. Visual observations of





exposed channel bedrock were not incorporated into the bedrock depth averages presented in
Table 1.

**Table 1.** *The median bedrock depth along with the elevation, mean, and 7-d maximum summer*
*temperatures over the period of record collected at most downstream site location in each*
*subwatershed.*

| site | 3D subwatershed surface area | mean valley width | median bedrock depth | most downstream stream temperature site | | |
|---|---|---|---|---|---|---|
| | | | | elevation | mean | 7-d max |
| | (km²) | (m) | (m) | (m) | (°C) | (°C) |
| Hughes River | 42.2 | 73.7 | 1.52 | 307 | 18.7 | 21.2 |
| Meadow Run | 15.0 | 55.3 | 1.93 | 450 | 18.4 | 20.4 |
| Jeremy's Run | 37.5 | 51.8 | 1.94 | 286 | 19.6 | 23.6 |
| Hazel River | 22.5 | 48.3 | 1.98 | 328 | 18.5 | 21.7 |
| Paine Run | 21.7 | 51.6 | 2.24 | 426 | 18.8 | 20.9 |
| Whiteoak Cyn. | 22.4 | 45.0 | 2.38 | 348 | 18.7 | 21.2 |
| Piney River | 20.6 | 48.6 | 2.54 | 371 | 17.9 | 20.6 |
| Staunton River | 18.0 | 45.6 | 3.43 | 309 | 17.4 | 19.9 |


*4.2 Spatial Dewatering Patterns and Climate Data*

Cumulative monthly precipitation during baseflow summer (July-September) was higher
than normal in 2016 and near average or lower than average (period of record 1942-2020),
depending on the month, in 2019 (Figure A2). Mean monthly air temperatures were higher than
average for both study years during baseflow summer reflecting the long-term trend of
increasing air temperatures in the park (Luray weather station GHCND:USC00445096; see
Menne et al. 2012). Patches of stream dewatering were observed along five of the eight study
subwatersheds between 19-27 July, 2016, when over 98 km of total stream length were mapped
(Figure 5). However, for Meadow Run, Hazel River, and Hughes River stream dewatering only
occurred near the upper stream origination point. In contrast, Paine Run and Jeremy's Run had
several discrete dewatering sections further from their origination points (examples shown in
Figure 3d, Figure A3). During the drier period 17-19 September 2019, no dewatering was found
along lower Staunton River, though Piney River had seven discrete dry patches where none were





mapped in 2016, and similar patterns were observed for those two streams in 2021 (Figure 6).
Paine Run had 29 points of dewatering in 2019, distributed mainly along the central and upper
sections of the stream corridor, and showed extensive dewatering in 2021 (Figures 5, 6, 7). The
two Paine locations that were dry in 2016 were also dry in 2019 and 2021.

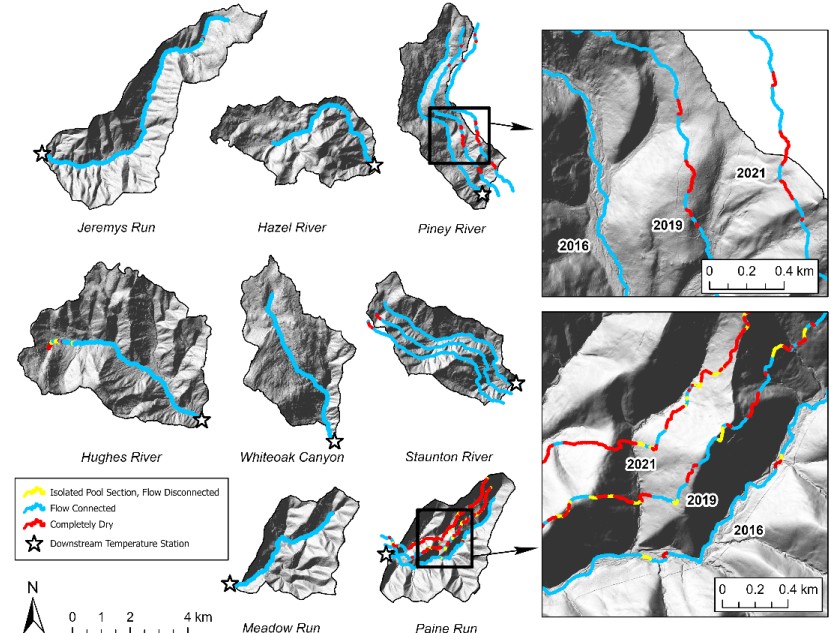

*Figure 5. Results from 2016, 2019, and 2021 longitudinal channel dewatering surveys conducted*
*by physical observation, where the 2019 and 2021 data are shown offset laterally from the*
*stream channel where those surveys occurred.*





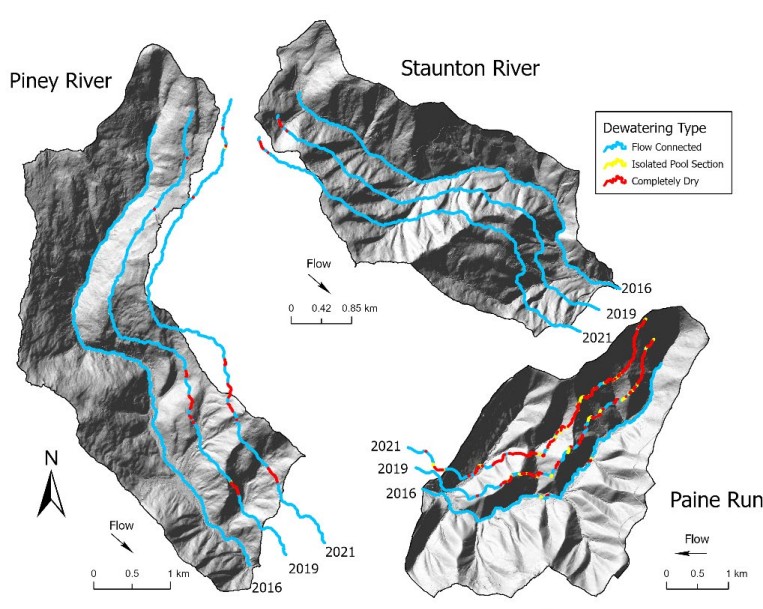

*Figure 6. Zoom views for the three subwatersheds where stream dewatering observations were*
*also collected in 2021.*



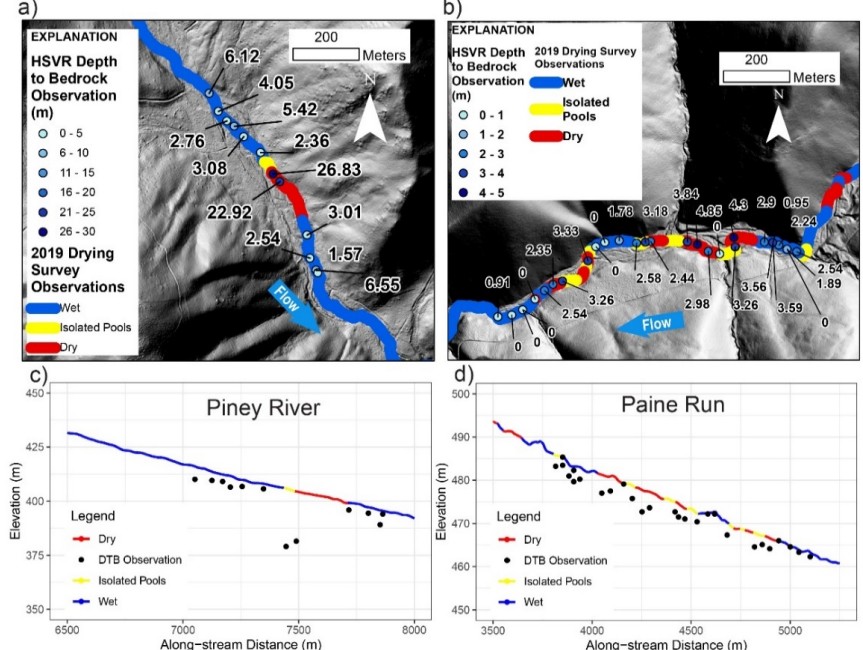

*Figure 7. The results of the 2019 stream drying survey and 2020 high spatial resolution HVSR*
*measurements are shown over the LiDAR hillshade in plan view (panels a, b) and along a*
*LiDAR-derived stream elevation profile cross-section view (panels c, d) for Piney River (panels*
*a, c) and Paine Run (panels b, d).*

*4.3 Stream Temperature Patterns*

Paired air and water annual temperature signals exhibited a spectrum of shallow

groundwater influences: phase shifts between stream and local air signals ranged from

approximately 5 to 30 d with a mean of 11 d. Reduced annual temperature signal amplitude ratio

generally corresponded with increased phase shift when all SNP stream monitoring sites are

plotted in aggregate (Figure 8a). Staunton River stream sites cluster together and show less

signal phase shift (mean of 10 d) for similar low amplitude ratio values (mean of 0.6) observed in

other subwatersheds.

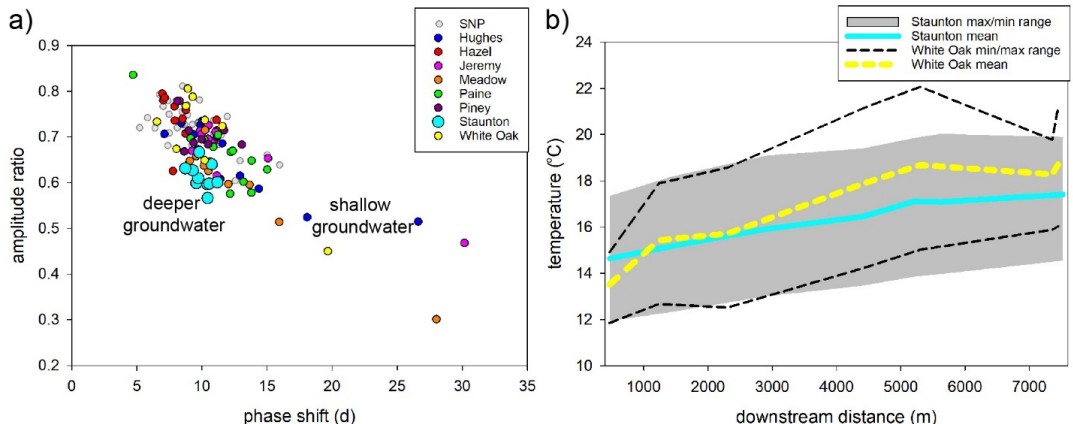

*Figure 8. Panel a) shows the annual temperature signal metrics for the study subwatersheds*
*highlighted within the larger SNP dataset with conceptual groundwater end member signature*
*trajectories. Panel b) displays the downstream mean summer temperature profiles and 7-d*
*maximum and minimum temperature ranges for Staunton River and Whiteoak Canyon.*

Although originating in a similar place, the downstream mean, 7-d maximum, and 7-d

minimum stream temperature profiles differed between Staunton River and Whiteoak Canyon,

where the latter had greater temperature variation and warming with downstream distance

(Figure 8b). The mean summer stream temperature had an approximate 2 °C total range over the

period of record. The warmest average (19.6 °C) and 7-d maximum (23.6 °C) was observed for

the lower Jeremy's Run site, which was also at the lowest elevation. However, only 23 m higher

in elevation, the downstream Staunton River site had the coldest average (17.4 °C) and 7-d

maximum (19.9 °C) summer temperature. Piney River, which has the second largest median

bedrock depth (2.54 m), had the second lowest average temperature (17.4 °C) at the lower site.

No significant relation was observed between elevation and mean summer temperature at the

lower stream monitoring site (Figure 9a), but a significant negative linear relation ($R^2$=0.52;

p<0.05) was determined between median stream corridor bedrock depth and mean summer

stream temperature (Figure 9b). However, there is strong leverage on the linear fit from the

Staunton River datapoint such that the Spearman rank test was not significant upon its removal
(r=-0.42; p=0.29).

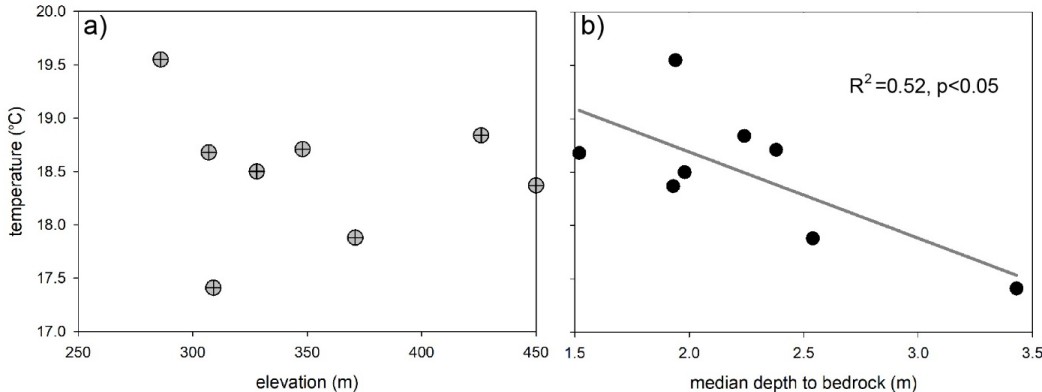

*Figure 9. Mean summer temperature at the downstream monitoring site is shown plotted by a)*
*elevation, and b) median subwatershed bedrock depth. A significant linear relation was*
*determined with bedrock depth but not elevation.*
*4.4 Baseflow Separation (Index)*

The summer season BFI determined for Paine Run, Piney River, and Staunton River over

show substantial variability, but the median summer BFI over the period of flow record for
Staunton River (0.62) is approximately 50% greater than Paine Run and Piney River (0.46 and
0.41, respectively, Table 2). For the primary study years of 2016-2019, Staunton River BFI is
always largest, and all sites are above their respective interquartile range in 2017 but below their
interquartile range in 2018 (Figure 10). The anomalously low 2018 BFI values can be explained
by extremely high summer precipitation that year (Figure S2), resulting in total streamflow being
dominated by runoff and quickflow as parsed with baseflow separation. Mean summer
streamflow over the period of record was highest for Piney River and lowest for Pain Run, and
overall summer streamflow was most stable for Staunton River (lowest coefficient of variation).
**Table 2.** The median summer Baseflow Index (BFI), mean summer streamflow, and mean summer
standard deviation (SD) streamflow for three gaged streams from 1993-2020.



| site | median BFI | mean streamflow (L/s) | mean coefficient of streamflow variation |
|---|---|---|---|
| Paine Run | 0.46 | 93.0 | 1.6 |
| Piney River | 0.41 | 164.4 | 1.7 |
| Staunton River | 0.62 | 157.3 | 0.7 |


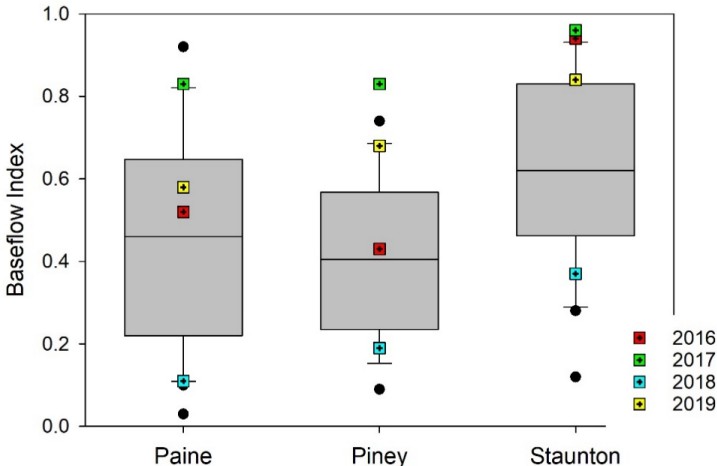

*Figure 10. Summer Baseflow Index metrics summarized from 1993-2020 for three streams, with*
*specific values from the primary study years identified.*
**5.0 Discussion**
*5.1 Longitudinal Spatial Structure in Observed Bedrock Depth*
Seminal groundwater/surface water exchange research has indicated that bedrock
topography along headwater streams may be a first-order control on the arrangement of nested
gaining and losing flowpaths (e.g. Tonina & Buffington, 2009), and increased (low permeability)
bedrock depth is recognized as a primary driver of stream disconnection during dry periods that
could be exacerbated by climate change (Ward et al 2020). However, despite the apparent
importance to a range of headwater stream physical processes and cold-water habitat, local



bedrock depth data are almost universally lacking, even in heavily studied experimental
watersheds. Our study provides new inferences regarding the effects of bedrock depth on stream
flow continuity, groundwater exchange, and temperature patterns in mountain streams. The
combined datasets indicate stream channel bedrock depth assessments may be necessary to
support stream habitat assessments and predictions of stream connectivity under drought and
climate change when existing large-scale geologic datasets are not of sufficient spatial resolution
to support natural resource management applications.

Bedrock depth varied substantially within and among several of the eight study SNP

subwatersheds but was predominantly shallow. For half of the subwatersheds (Hughest River,
Meadow Run, Jeremy's Run, and Hazel River), median bedrock depth along the stream channel
and lateral riparian zone was less than 2 m and did not show notable variability with distance,
outside of one 12.8 m depth to rock location at upper Hazel River (Figure 4). This anomalous
measurement at Hazel was collected lateral to the stream on a valley terrace of colluvium, in the
vicinity of the only cold (approximately 10 °C at land surface) riparian spring that was observed
during all HVSR surveys. Bedrock depths of greater than 8 m were found along the upper
Whiteoak Canyon riparian zone as well (Briggs et al 2018a), also associated with surficial
seepage. Two anomalous bedrock depth measurements of 22.9 and 26.8 m were collected along
the Piney River channel, but instead of being associated with groundwater springs, they
coincided with a discrete sections of channel dewatering at baseflow during 2019 and 2021.
Therefore, it appears that discrete zones of thick surficial material are the exception along SNP
streams, though they can be important to localized processes such as focused riparian discharge
and streamflow disconnection (latter discussed in Section 5.2).



There are several existing sources of bedrock depth data that could potentially be used to
inform headwater stream modeling and habitat assessment, but the accuracy of such datasets
along headwater streams (typically away from existing boreholes) has generally not been
evaluated. We conducted a point-scale comparison of our relatively high-resolution bedrock
depth measurements to the global bedrock depth map of Shangguan et al., (2017) and found that
bedrock depths were almost universally overpredicted at the SNP by large margins (Figure A4).
Specifically, predictions from the global-scale dataset exceeded HSVR measured depths by
+12.2 m (mean), or approximately four times the average bedrock depth (2.9 m). As baseflow
generation is expected to be dominated by shallow groundwater sourced from unconsolidated
sediment in these headwater systems, this differential could propagate substantial uncertainty
into process-based groundwater flow model predictions if the global-scale dataset was used to
inform model structure.
Publicly available maps of surficial geologic materials are another potential source of
bedrock depth information. High-resolution digital soils maps are now widely available,
including for the catchments of SNP, and these maps do capture some of the general depth to
rock transitions between subwatersheds observed in this study. For example, NRCS (2020)
(https://websoilsurvey.sc.egov.usda.gov/App/WebSoilSurvey.aspx, accessed 12/10/2020)
indicate that the Whiteoak Canyon stream corridor is comprised of silts, loams, and stony soils
with a general bedrock depth of approximately 1.2 m., which is in a similar range as most HVSR
measurements made along the upper stream section (Figure 4). However, the generalized soil
units may not offer needed detail regarding site-specific valley sediment thickness for
hydrogeological and ecological studies where information regarding within-watershed variation
is critical. Along lower Piney River, where HVSR data had depths to rock ranging 1.4 to 3.6 m,

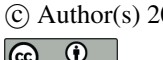

the NRCS soils map universally indicates silt and stony material > 2 m. Along Paine Run, where
the stream is often scoured to bedrock, the soils map shows consistent highly permeable sandy
material with > 2 m thickness. This discrepancy is understandable given most of the test pits
were likely substantially further downstream in better terrain for agriculture. In conclusion,
analysis of large-scale patterns from existing soils maps and interpolated/predicted bedrock
depth layers indicates that more precise geophysical mapping of bedrock depth may be needed to
inform stream research and management, particularly in shallow, low-permeability bedrock
terrain.
*5.2 Summer Stream Dewatering Related to Bedrock depth*

Aligned with the conceptual model of Ward et al., (2018), our central hypothesis was

bedrock depth along the stream corridor acts as a primary control on longitudinal stream
dewatering and flow disconnection during summer low flows (visual example shown in Figure
A3). We postulated that permeable streambed thickness may undulate along mountain stream
channels, and relatively thick sub-stream sediment zones could accommodate the entirety of low
streamflow volumes, locally disconnecting channels during seasonal drydown. We found mixed
support for this simple hypothesis. Hazel River and Hughes River were two of the three
subwatersheds that had dry channel zones just downstream of their respective stream origination
points in 2016, and these two riparian corridors also had their deepest riparian bedrock depths in
those high-elevation areas. However, as discussed above, Whiteoak Canyon had relatively thick,
porous sediment zone near the subwatershed outlet but did not show any zones of dewatering,
nor did lower Staunton River in 2016, 2019, or 2021, despite having the deepest median bedrock
contact. Jeremy's Run had three mapped dry zones in 2016 (not surveyed in 2019), yet depth to
rock in those areas was only approximately 2 m, though the HVSR data collection points were
not perfectly aligned with the dry patches. To address this spatial mismatch in stream dewatering





and HVSR data, we used the stream dewatering maps to guide two new high-resolution HVSR
surveys in March 2020 along sections of Paine Run and Piney River with dynamic patterns of
channel drying, as described below.

When bedrock depth data were collected at high-resolution, even more variability in

bedrock topography/sediment thickness was revealed then in the original larger-scale surveys,
and that finer scale of information was relevant to understanding stream dewatering patterns.
For example, during summer 2019, a 291 m length section of lower Piney River was observed to
be dry, and immediately preceded by 62 m of isolated stream channel pools, and a nearly
identical dewatering pattern was observed there in 2021 (Figures 6,7a,c). The upper portion of
this major feature of stream disconnection corresponded directly with a transition in bedrock
depth along the channel from approximately 3 m to adjacent measurements of 27 and 23 m. This
'trough' in the bedrock surface can likely act as a streamwater sink (shown conceptually in
Figure 1b), routing surface water downward to the point of draining the channel locally in the
summers of 2019 and 2021, but not in 2016 when precipitation (groundwater supply) was higher
than normal. Further downstream, the bedrock depth returned to approximately 3 m near the
furthest downstream measurement point, and flowing channel water was again noted during the
drying surveys. Such a section of stream dewatering in the lower watershed would serve to
impede fish passage along Piney River during the lowest flows, likely corresponding to times of
maximum thermal stress when fish mobility is critical to seeking thermal refuge (Magoulick and
Kobza, 2003).

Not all variability in bedrock depth below streams associated with stream drying was as

dramatic as the Piney River example but can be important in disconnecting channel habitat in
summer. Paine Run is a more strongly confined stream valley that had 29 discrete zones of

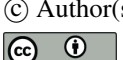



stream channel dewatering during September of 2019 and extensive dewatering in 2021, when
numerous dead brook trout were also noted (Figures 5,6,7b,d). Paine also had the greatest total
exposed bedrock out of any of the SNP subwatersheds in this study, indicating a highly
constrained valley underflow reservoir. High resolution bedrock depth data was collected over a
Paine Run subreach with seven discrete dry patches ranging from 17 m to 185 m in channel
length, with many bordered by zones of isolated pools (Figure 7b). A comparison of these
patterns with bedrock depth along the channel shows the flowing sections of stream were
dominated by exposed bedrock surfaces or thin sediment. However, a notable exception is
toward the upstream end of this focus reach, where depth to rock was consistently > 2 m over the
run up to a large zone of disconnected channel with some isolated pools (Figure 7b,d). This
result suggests the losses of stream water accumulated over this approximately 80 m channel
distance. In the following downstream contiguous sections of dry channel and/or isolated pools,
bedrock depth averaged a larger 3.3 m, indicating the entirety of streamflow was accommodated
by the subsurface, congruent with our original hypothesis. However, knowledge of bedrock
depth in isolation is clearly not sufficient to predict stream channel gaining, losing, and
disconnection patterns as the stream with the largest average bedrock depth, lower Staunton
River (median depth to rock 3.4 m, Figure 4), was not observed to dewater during any of the
three physical surveys (Figures 5,6).
*5.3 Summer Stream Temperature and Groundwater Exchange Dynamics*
Although headwater stream heat budgets are complex, our data indicates groundwater

connectivity plays an important role when stream temperatures are already close to aquatic
species thermal tolerances. The apparent dominance of shallow (<3 m depth) groundwater
discharge along Whiteoak Canyon contributed to the Briggs et al. (2018b) prediction that the
lower reaches would not provide suitable brook trout habitat by the end of the century given


anticipated atmospheric warming. Jeremy's Run, a long (13.4 km) stream consistently underlain
by a shallow bedrock contact (median depth < 2 m), already shows a 7-d maximum summer
temperature that exceeds expected brook trout tolerances (i.e., >23.3 °C mean weekly average
temperature, Wehrly et al., (2007)) along the lowest reach.

The underflow reservoir of headwater stream valleys integrates upgradient and lateral

hillslope groundwater flowpaths, which accumulate with distance when bounded by low
permeability bedrock. The two subwatersheds with largest median bedrock depth along their
respective upstream corridors had the coldest mean summer temperatures, with Staunton River
standing out as distinctly colder, and having the only 7-d max temperature below 20 °C (Table
1). There was a significant relation between median bedrock depth and mean summer stream
temperature at the lower stream sites but not with elevation (Figure 9), indicating exchange with
groundwater had disrupted the expected elevation control on lower reach cold water habitat.
Surficial hillslope contributing area is often assumed a primary control on potential groundwater
discharge at the stream subreach scale. However, Staunton River also had the second smallest
drainage surface area of all study subwatersheds, and it is often assumed that lateral groundwater
inflow to headwater streams is related to presumed upslope contributing area. Further, Staunton
River did not have an average valley bottom width that was greater than other streams that were
observed to dewater.

Our research indicates that the vertical shallow aquifer dimension, as represented by

bedrock depth, is likely an important control of groundwater storage and connectivity to the
stream corridor. This conclusion is supported by the paired air/water annual temperature signal
metrics, indicating Staunton River sites cluster in the stronger, deeper groundwater influence
compared to most observations along the other SNP streams (Figure 8a). Therefore, it seems





there are important tradeoffs between bedrock depth along the stream channel as a driver of
stream dewatering and sediment thickness along the valley floor and hillslopes as a potential
source of stream baseflow.
For a more in-depth analysis the paired bedrock depth and groundwater inflow controls
on headwater summer stream dynamics, Staunton River can be contrasted with Paine Run. The
latter had a similar drainage surface area to Staunton River, but a 1.2 m shallower bedrock depth
on average, showed dozens of dewatered stream channel sections in 2019 and 2021, and had a
downstream boundary summer stream temperature that was 1.4 °C warmer. In addition to a
reduced average bedrock depth, Paine Run had numerous sections of exposed bedrock adjacent
to localized pockets of stream channel alluvium and colluvium (Figure 4, 7), while extensive
colluvial deposits along the Staunton River channel limited exposed bedrock to a few m-scale
sections associated with pool steps (Figure 4). Lower Staunton experienced major debris flows in
June, 1995 (Morgan and Wieczorek, 1996), events that likely created an enhanced local
groundwater reservoir within coarse hillslope material compared to other SNP subwatersheds.
Based on the integrated datasets from these two SNP streams, we conclude that
groundwater exchange is a critical factor determining whether headwater streams will warm and
dewater in summer, which in turn is controlled in part by the thickness of supra-bedrock
unconsolidated aquifer. As noted above, annual temperature metrics indicated a consistently
deeper groundwater discharge influence along Staunton River, while Paine Run had annual
signal metrics that mainly indicated reduced and/or more shallow groundwater influence (Figure
9a). Long term streamflow and baseflow analysis from these streams showed Staunton River had
higher, but more stable summer discharge (Table 1), and substantially higher median summer
BFI (0.62 vs 0.46), indicating greater dominance of groundwater as a generator of streamflow



compared to runoff and quickflow. Previous research in SNP used paired air/stream water
temperature records, precipitation, and landscape characteristics to statistically model
'groundwater influence' by year on a scale of 0-1 at the 100-m scale along the streams of this
study, where details are described by Johnson et al., (2017). Although this previous work only
extended to 2015, that year had analogous BFI scores to 2019 for Staunton River (0.88 vs 0.84)
and Paine Run (0.60 vs 0.58). Comparing the 2019 drying survey observations to the 2015 high
spatial resolution modeling of groundwater influence we found that Paine Run was predicted to
have groundwater influenced tributaries, but along the mainstem, where extensive dewatering
was observed, there was substantially reduced modeled groundwater influence compared to the
mainstem of Staunton River (Figure 11).



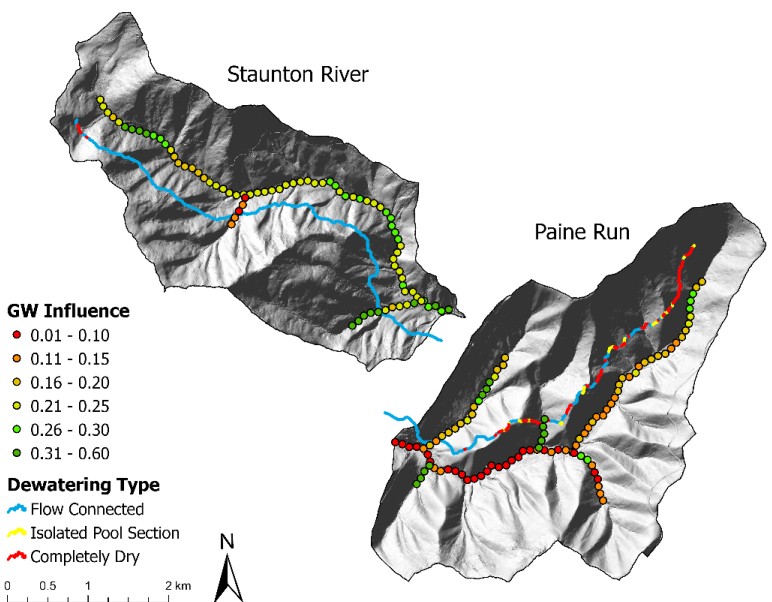

*Figure 11. The 2019 stream dewatering survey data (lines; this study), plotted offset of the*
*mainstem, and 100-m groundwater influence predictions (points from Johnson et al., 2017),*
*plotted along the mainstem and tributaries, of Staunton River and Paine Run.*
*This observation and model comparison represents another line of evidence that groundwater*

connectivity at the sub-reach scale is key in determining whether local increases in depth to

bedrock drive channel dewatering at low flow. The impact of reduced underflow groundwater

supply on stream disconnection is likely exasperated by the extensive zones of exposed bedrock

along Paine Run (Figure 4, 7d), which locally reduce groundwater mounding in stream valley

sediments as shown conceptually in Figure 1b, such that abrupt increases in bedrock depth cause

stream dewatering. Among the eight streams investigated here, Staunton River likely represents

the most resilient summer cold water habitat, which could not be predicted using bedrock depth

data alone but necessitated paired assessment of groundwater discharge dynamics.



## 6 Conclusions


In steep mountain valley stream systems underlain by low-permeability bedrock, the
longitudinal underflow reservoir serves as a complex mechanism of streamflow generation,
streamflow losses, and stream temperature control (Figure 1, Supplementary Figure S1). Our
study utilized complimentary geophysical, temperature, and hydrologic data at the scale of eight
subwatersheds to highlight apparent tradeoffs in bedrock depth, shallow groundwater supply, and
the quality of cold-water habitat. Certain mountain stream corridor parameters may be
reasonable to assume or infer from high-resolution topographic data, such surficial sediment
permeability (based on land surface roughness) and stream valley width, which are primary
controls on whether underflow serves as a net source or sink of stream water (Flinchum et al.,
2018; Ward et al., 2018). However, as shown here, advances in predicting hydrologic
connectivity and thermal variation along mountain stream networks may also require local
evaluation of bedrock depth and stream-groundwater exchange.
When local increases in bedrock depth are not balanced by groundwater inflow, streams
may be expected to dewater and disconnect under low flow conditions, and streams with reduced
deeper groundwater influence show warmer summer temperatures. Contrary to what might be
expected, we found that mean summer stream temperature at was not significantly related to
elevation at all lower study boundaries, but instead was (negatively) related to average stream
bedrock depth. Staunton River had the coldest summer stream temperatures and most
pronounced deeper groundwater signatures. However, that subwatershed was of relatively small
total surface area and average valley width. The defining physical feature of Staunton River was
that it had the largest average bedrock depth of all the eight SNP study streams at 3.4 m,
allowing greater overall storage of recharge and baseflow generation. The other two gaged



streams had substantially reduced baseflow indices, indicating streamflow generation was
dominated by runoff and quickflow.
Overall, SNP streams tended to have consistently shallow bedrock depth, though a subset
were more variable or had spatial trends and discrete features. Observed channel dewatering
patterns during late summer baseflow periods were related to local scale variation in bedrock
depth, such as a discrete feature of greater than 20 m depth observed along Piney River that
caused repeated streamflow disconnection. However, in other streams more subtle bedrock depth
variation also caused channel dewatering, indicating the importance of local hydrogeological
context in determining the importance of bedrock depth on streamflow connectivity. For
example, patchy 2-4 m deposits of sediment adjacent to exposed bedrock along Paine Run
caused extensive summer dewatering in 2019 and 2021, and during the latter survey many dead
brook trout were noted in the disconnected sections. Paine and Piney also showed enhanced
dewatering during the summers of 2019  and 2021 compared to the wetter 2016 summer,
demonstrating the additional control of recent precipitation on stream disconnection in headwater
systems that do not efficiently store water.
Lateral groundwater inflow through high permeability, unconsolidated sediments is a
critical component of headwater stream baseflow (Tran et al., 2020). Shallow, low permeability
bedrock can constrain lateral flowpaths and underflow to the near surface critical zone, where it
is highly sensitive to enhanced evapotranspiration, temperature increase, and drought under
climate change (Condon et al., 2020; Hare et al., 2021). As it becomes increasingly important to
understand and predict the resilience of mountain cold-water stream habitat at a fine spatial
grain, continued coupled advances in geophysical characterization, stream temperature
monitoring, and groundwater exchange analysis are needed.





**Data Availability**

The data described in this manuscript are available at: doi.org/10.5066/F7B56H72,

doi.org/10.5066/F7JW8C04, and doi.org/10.5066/P9IJMGIB

**Author contribution**

*Conceptualization*: M.A. Briggs, Z.C. Johnson, C.D. Snyder, N.P. Hitt; *Investigation*: M.A. Briggs, P. Goodling, Z.C. Johnson, C.D. Snyder, K.M. Rogers, N.P. Hitt; *Visualization*: M.A. Briggs, K.M. Rogers, P. Goodling, J.B. Fair, C.D. Snyder. All authors contributed to the formal analysis varied stages of writing.

**Competing interests**

The authors declare that they have no conflict of interest.

**Acknowledgments**

The authors gratefully acknowledge support from Natural Resource Preservation Program and

the U.S. Geological Survey (USGS) Chesapeake Bay Priority Ecosystems Science and Fisheries

Program. We also thank the Shenandoah National Park Staff for site access and general support

and field support from John Lane, David Nelms, Adam Haynes, Erin Snook, David Weller, Evan

Rodway, Jacob Roach, Matt Marshall, Joe Dehnert, and Mary Mandt. Any use of trade, firm, or

product names is for descriptive purposes only and does not imply endorsement by the U.S.

Government.



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




**Appendix A**

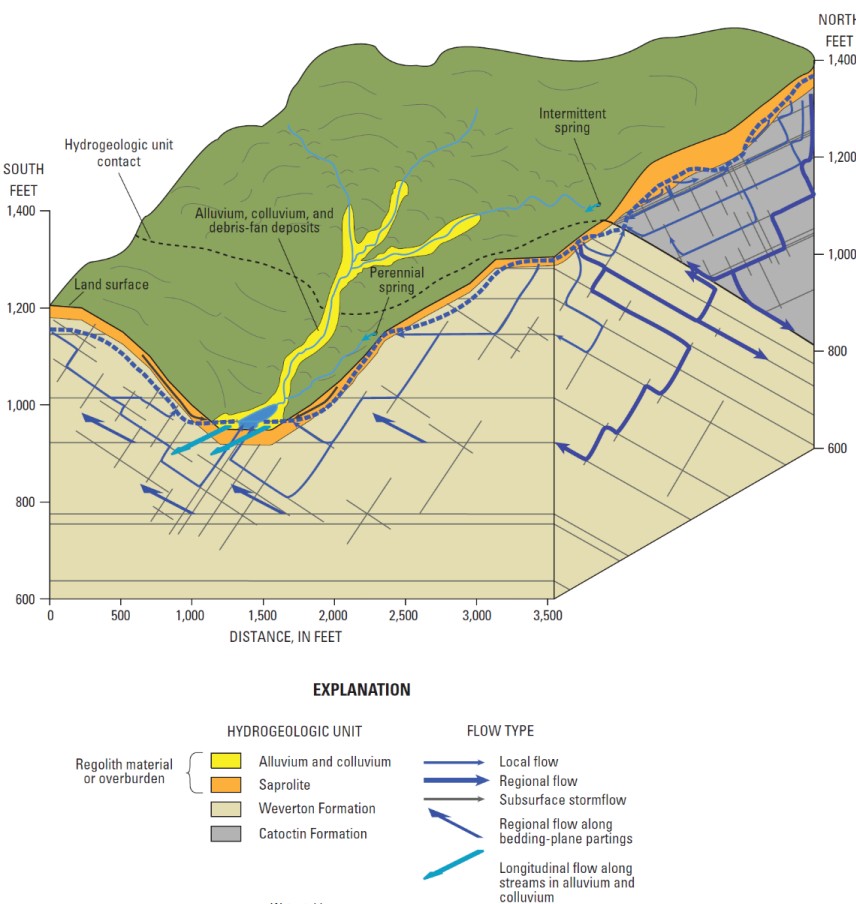

Figure A1. The headwater streams of Shenandoah National Park, Virginia, USA are expected to
flow over coarse alluvium and colluvium and have connectivity to shallow hillslope groundwater
and underflow, but reduced connectivity to deeper bedrock groundwater (Modified Figure 26 in
(Nelms and Moberg, 2010) *U.S. Geol. Surv. Investigations Rep.* 2010–5190.

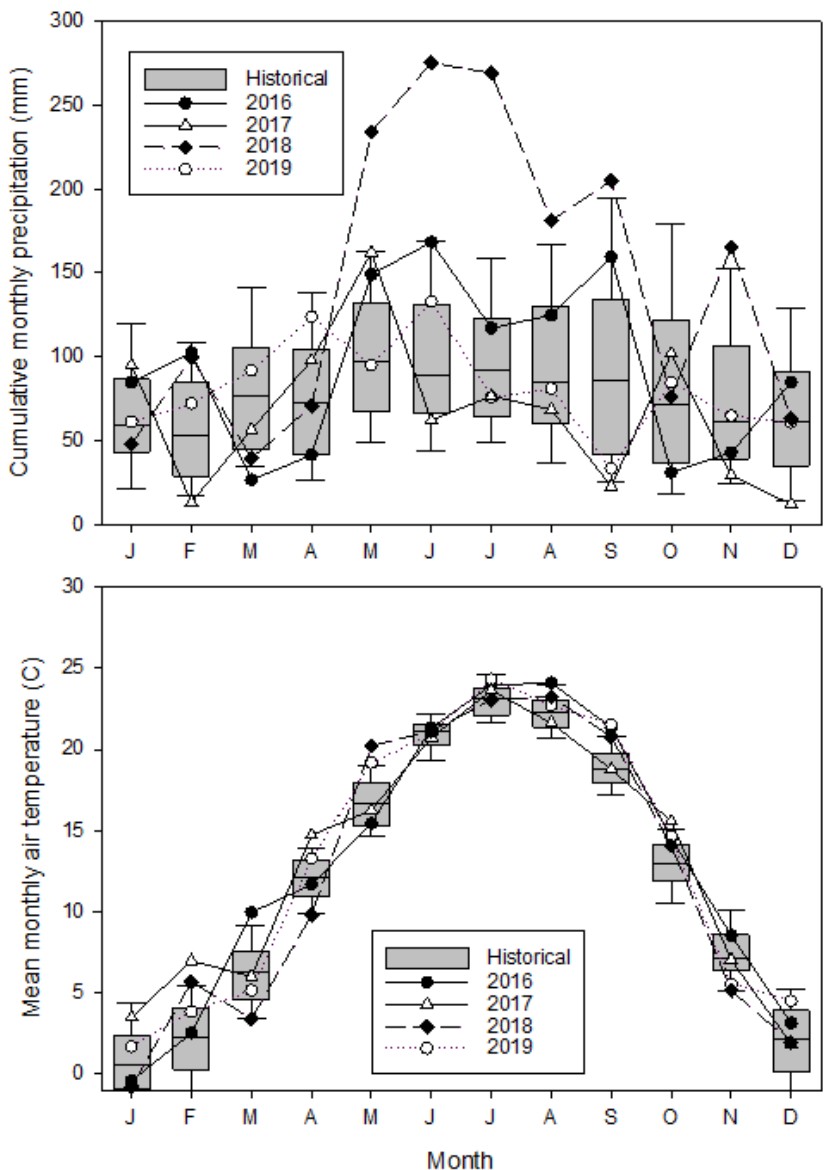

Figure A2. Monthly precipitation and air temperature data derived from the Luray weather
station (GHCND:USC00445096) located within Shenandoah National Park. Box plots show the
distribution of values for the period of record (1942-2020) with the limits of the box containing
50% of the values, whiskers containing 90% of the values, and solid line in boxes depicting the
median value. The lines represent values for the four primary study years.



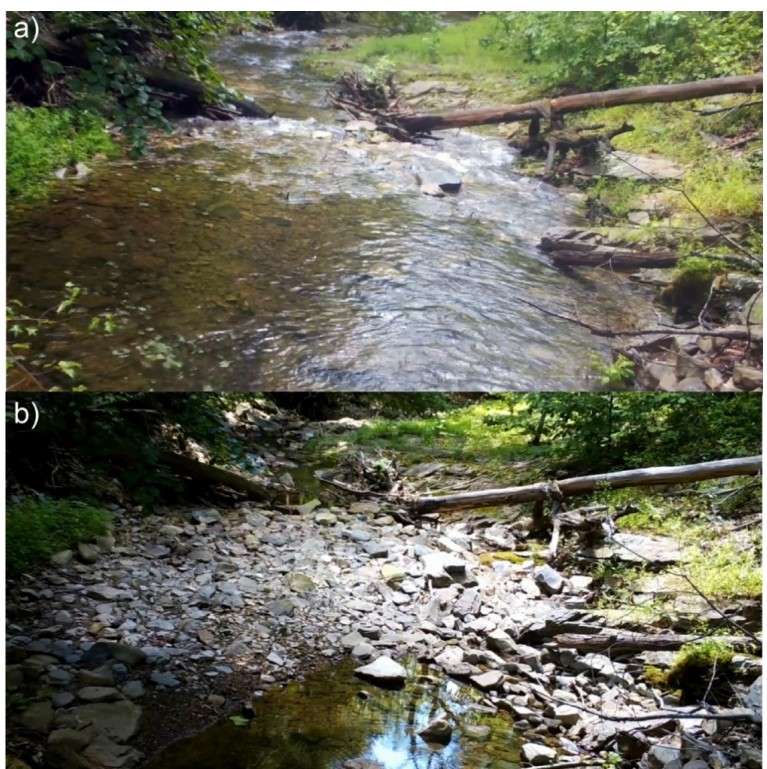

Figure A3. Images from the same vantage point along Paine Run during a) high and b) low flow
times, the latter showing channel dewatering associated with a deposit of coarse alluvium across
the channel.



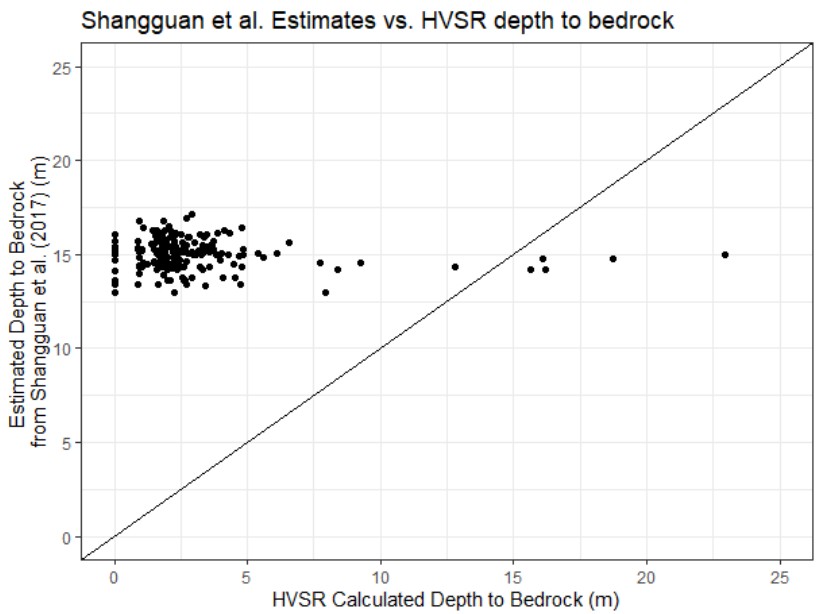

Figure A4: Comparison between bedrock depth modeled for the globe by Shangguan et al.,
(2017) at a 250m resolution and the HVSR-calculated depths to bedrock in this study.

**Appendix B**

Table B1. Summer stream temperature metrics for each study subwatershed determined from the
data set of Snyder et al. (2017), doi.org/10.5066/F7B56H72.

| Subwatershed | SiteID | Easting | Northing | Downstream Distance (m) | summer mean (°C) | 7-d min (°C) | 7-d max (°C) | Stdev (°C) |
|---|---|---|---|---|---|---|---|---|
| Hughes | HUR1MP | 730038 | 4276000 | 242.50 | 13.43 | 11.50 | 16.07 | 0.98 |
| Hughes | HUR3LCP | 731058 | 4275970 | 1634.44 | 15.73 | 13.23 | 18.21 | 1.24 |
| Hughes | HUR5LCP | 732278 | 4275850 | 3308.93 | 16.21 | 13.48 | 18.62 | 1.24 |
| Hughes | HUR6MP | 733348 | 4275060 | 5163.46 | 16.39 | 13.86 | 17.95 | 1.13 |
| Hughes | HUR12MP | 733698 | 4274880 | 5620.73 | 16.79 | 14.09 | 18.34 | 1.24 |
| Hughes | HUR8LCP | 733988 | 4274619 | 6219.50 | 18.80 | 15.19 | 21.49 | 1.57 |
| Hughes | HUR9LCP | 733968 | 4274529 | 6284.35 | 17.59 | 14.49 | 20.09 | 1.34 |
| Hughes | HUR10MP | 734928 | 4273520 | 8187.04 | 18.05 | 15.01 | 20.05 | 1.40 |
| Hughes | HUR13MP | 735258 | 4273330 | 8667.16 | 18.68 | 15.15 | 21.18 | 1.62 |
| Hazel | HZR1MP | 735158 | 4278560 | 707.45 | 16.80 | 13.26 | 19.75 | 1.46 |
| Hazel | HZR3LCP | 735498 | 4278760 | 1190.18 | 16.74 | 13.08 | 19.41 | 1.50 |
| Hazel | HZR11MP | 736378 | 4279640 | 2951.89 | 18.16 | 15.34 | 20.32 | 1.52 |





| Hazel | HZR5MP | 736638 | 4279790 | 3331.66 | 17.59 | 13.59 | 20.62 | 1.67 |
| Hazel | HZR6MP | 737498 | 4279059 | 5095.01 | 18.16 | 14.03 | 21.40 | 1.74 |
| Hazel | HZR7MP | 738048 | 4277990 | 6820.13 | 18.48 | 14.50 | 21.77 | 1.72 |
| Hazel | HZR9MP | 738368 | 4277620 | 7478.33 | 18.50 | 14.74 | 21.72 | 1.63 |
| Jeremy's | JR1MP | 734618 | 4293430 | 102.97 | 15.48 | 12.42 | 18.74 | 1.41 |
| Jeremy's | JR2MP | 733908 | 4293130 | 1268.08 | 16.49 | 13.38 | 19.42 | 1.38 |
| Jeremy's | JR4MP | 732498 | 4292250 | 3699.53 | 16.84 | 14.23 | 18.22 | 1.11 |
| Jeremy's | JR5MP | 731778 | 4290670 | 5961.22 | 17.54 | 14.15 | 20.24 | 1.43 |
| Jeremy's | JR13MP | 731498 | 4289490 | 7506.87 | 18.16 | 14.57 | 21.28 | 1.67 |
| Jeremy's | JR7MP | 730068 | 4288080 | 10327.83 | 18.76 | 16.79 | 21.07 | 1.10 |
| Jeremy's | JR9LCP | 729888 | 4288080 | 10539.49 | 17.78 | 15.33 | 20.01 | 1.04 |
| Jeremy's | JR12MP | 728758 | 4288080 | 12030.09 | 18.61 | 14.46 | 22.08 | 1.73 |
| Jeremy's | JR10MP | 727758 | 4288440 | 13376.47 | 19.55 | 14.93 | 23.55 | 1.98 |
| Meadow | MR0MP | 695318 | 4228150 | 0.00 | 14.16 | 12.01 | 15.57 | 1.07 |
| Meadow | MR1MP | 695038 | 4227980 | 217.46 | 16.82 | 13.82 | 18.39 | 1.28 |
| Meadow | MR2MP | 694678 | 4227520 | 979.43 | 17.11 | 13.71 | 18.78 | 1.48 |
| Meadow | MR9MP | 693488 | 4227270 | 2757.87 | 18.10 | 15.34 | 19.71 | 1.31 |
| Meadow | MR4LCP | 693428 | 4227240 | 2854.69 | 17.01 | 13.92 | 19.02 | 1.44 |
| Meadow | MR8MP | 693078 | 4226450 | 4036.50 | 17.53 | 14.32 | 19.48 | 1.35 |
| Meadow | MR6LCP | 692918 | 4226170 | 4446.20 | 17.08 | 14.34 | 19.32 | 1.29 |
| Meadow | MR7MP | 691738 | 4225700 | 6209.68 | 18.37 | 15.33 | 20.44 | 1.44 |
| Paine | PAR1MP | 696938 | 4232031 | 249.71 | 16.86 | 13.96 | 18.72 | 1.36 |
| Paine | PARB1 | 696718 | 4231390 | 1115.08 | 17.20 | 14.81 | 18.70 | 1.15 |
| Paine | PAR2MP | 696468 | 4231210 | 1542.16 | 17.15 | 15.22 | 18.61 | 1.03 |
| Paine | PAR3MP | 695685 | 4230400 | 3169.18 | 17.48 | 14.93 | 19.28 | 1.15 |
| Paine | PAR5LCP | 695369 | 4230040 | 3861.10 | 17.87 | 15.01 | 19.53 | 1.32 |
| Paine | PAR9MP | 694568 | 4229850 | 5016.00 | 18.04 | 15.43 | 19.60 | 1.12 |
| Paine | PAR6MP | 694218 | 4229700 | 5563.29 | 18.39 | 14.86 | 20.32 | 1.52 |
| Paine | PAR10MP | 694068 | 4229730 | 5829.23 | 18.62 | 14.71 | 20.50 | 1.65 |
| Paine | PARB2 | 693248 | 4230140 | 7055.48 | 18.60 | 14.54 | 20.57 | 1.67 |
| Paine | PAR8MP | 693137 | 4230180 | 7122.47 | 18.84 | 14.50 | 20.91 | 1.97 |
| Piney | PIR1MP | 736308 | 4292604 | 402.61 | 15.67 | 12.24 | 19.16 | 1.65 |
| Piney | PIR3LCP | 736218 | 4291980 | 1199.93 | 16.43 | 12.76 | 19.78 | 1.65 |
| Piney | PIR4MP | 735598 | 4291160 | 2480.47 | 16.55 | 13.24 | 19.65 | 1.51 |
| Piney | PIR5MP | 735458 | 4290050 | 3955.00 | 16.82 | 13.62 | 19.97 | 1.48 |
| Piney | PIR6MP | 736408 | 4289180 | 5862.79 | 17.74 | 15.87 | 20.49 | 1.15 |
| Piney | PIR7MP | 736748 | 4288300 | 7115.97 | 17.40 | 15.07 | 20.22 | 1.17 |
| Piney | PIR8MP | 737538 | 4287390 | 8756.79 | 17.88 | 14.63 | 20.55 | 1.39 |
| Staunton | SR1MP | 725248 | 4260810 | 477.07 | 14.64 | 11.96 | 17.33 | 1.32 |
| Staunton | SR2MP | 725908 | 4260450 | 1412.13 | 15.16 | 12.34 | 18.15 | 1.44 |
| Staunton | SR5MP | 726948 | 4259890 | 2907.57 | 15.92 | 13.03 | 19.08 | 1.51 |





| Staunton | SR6MP | 728018 | 4259921 | 4398.87 | 16.45 | 13.48 | 19.38 | 1.48 |
|----------|-------|--------|---------|---------|-------|-------|-------|------|
| Staunton | SR10MP | 728598 | 4259660 | 5220.08 | 17.12 | 13.88 | 19.84 | 1.48 |
| Staunton | SR7MP | 728718 | 4259390 | 5627.21 | 17.09 | 13.99 | 20.02 | 1.52 |
| Staunton | SR9MP | 729448 | 4258420 | 7519.72 | 17.41 | 14.57 | 19.88 | 1.33 |
| White Oak | WOC1MP | 728788 | 4273701 | 469.03 | 13.51 | 11.85 | 14.91 | 0.74 |
| White Oak | WOC3MP | 728998 | 4273160 | 1237.37 | 15.43 | 12.67 | 17.90 | 1.29 |
| White Oak | WOC4MP | 729268 | 4272400 | 2307.96 | 15.71 | 12.52 | 18.58 | 1.48 |
| White Oak | WOC5MP | 730288 | 4271180 | 4428.05 | 17.90 | 14.23 | 21.16 | 1.77 |
| White Oak | WOC7LCP | 730758 | 4270690 | 5302.94 | 18.69 | 15.02 | 22.07 | 1.79 |
| White Oak | WOC8MP | 730948 | 4269150 | 7356.87 | 18.29 | 15.88 | 19.78 | 1.07 |
| White Oak | WOCB | 731018 | 4269110 | 7448.09 | 18.71 | 16.04 | 21.18 | 1.28 |










