# Peer review of "Bedrock depth influences spatial patterns of summer baseflow, temperature, and flow disconnection for mountainous headwater streams"

_Hydrology and Earth System Sciences, 2021_

## Referee Comment (RC1)

Briggs *et al.* (HESS) review

First, thank you for the opportunity to review this paper. I really enjoyed reading it. In their paper, Briggs *et al.* collated geophysical surveys, remote sensing data, and stream temperature and discharge loggers to reveal the role of bedrock depth, in catchments underlain by low hydraulic conductance bedrock, on stream dewatering and thermal resilience. The authors have done an excellent job in highlighting the role of fine-scale hydrogeological setting on the aforementioned hydrological processes. Further, the authors revealed that global scale datasets of depth to bedrock (DTB) largely overestimate this critical parameter, by as much as 12 m. Ultimately, this piece is very timely, and adds a nice story to the hydrology puzzle. I especially applaud the authors in identifying some complex processes that really drive home the importance of surface water - groundwater interactions.

I am happy to recommend this paper for publication after what I consider minor to moderate revisions. This dataset is incredibly rich, and whilst I understand it is not possible to do everything, I do think there is some bandwidth for the authors to dig a bit deeper into what is at play in Staunton. I think given the density of these data, a few things may be conceptualized. For instance, wider valleys will have high solar loading, and the recharge will be spread over a larger area. What role might this have on thermal and discharge regimes?

Thanks again, and nice work!
Antóin O'Sullivan

Apologies for grammatical errors herein.

**Specific comments:**

L 134: True, however, could this also be a function of bedrock $K$? For bedrock with a relatively high $K$ (karst for instance), a duality may exist where a portion of the water is driven laterally - as stated - whilst another portion may be recharging the bedrock aquifer. These mechanisms are also likely temporally dynamic. In the setting of this study, I agree that lateral flow with bedrock shallowing seems most likely given low $K$. However, in the introduction, it may be best to speak exclusively to the conceptual controls in general.

L 162: Not sure if it is worthwhile stating consolidated sediments, e.g., clay, may dampen the signal too (see Haefner, R.J., Sheets, R.A. and Andrews, R.E., 2010. Evaluation of the horizontal-to-vertical spectral ratio (HVSR) seismic method to determine sediment thickness in the vicinity of the South Well Field, Franklin County, OH.). If one assumes macro-pores in the clay are limited, this may also lead to low storage capacity in areas of shallow depth to low hydraulic conductance layer?

L 187: this is an awesome study site overview. nice job!

L232: maybe spell out the acronym here as it is the first time it appears in a caption.

L244: nice study sites!

L253: this progression seems logical

L270: typo "bedrock". This aligns to my prior comment on the assumption the signal to being changed by bedrock. Hardpan or clay may also do this? This does not distract from the study, just worth a nod to the potential limitations of passive stratigraphy mapping

L 271: and here is the answer, awesome :) Could there be an *n* shown for number of these boreholes? Looks to be *n*=6 from Goodling et al. report?

L 290: typo 'teams'

L301: nice!

L329: are these gauges co-located with temperature sensors as displayed in Figure 5?

Figure 4 – this is a serious dataset, folks. Nice study design!

L352: smallest seems like an odd word for describing DTB, maybe lowest? This is not a game-changer, just a style thing.

Table 1 - a quick regression of valley width to median bedrock depth illustrates a power law relationship ($R^2$ ~0.62). Given how poor the broad scale depth to bedrock maps were at predicting bedrock depth, it may be useful to illustrate here that using high res LiDAR and valley morphology as controls on bedrock rock may provide a more realistic view of bedrock depth in these areas. See Figure 1 below. Something for appendix maybe, but I think you have shown in a remarkably clear way that we need better geophysical data. Awesome stuff.

Figure 5 – nice

L386 – might rephrase this sentence for clarity.

Figure 7 - this is a powerful figure. It brings a lot of questions to mind. I wonder how this would look if one added another 2 panels that plotted the same dewatering observations and sub DTB with valley width? The reason I suggest this, the valley is 3D, by accounting for this 3D space, and given the authors have this amazing data set, it may point towards a more robust understanding of x,y,z space on these hydro processes. See Figures 2 and 3 below.

L 435 – nice

L482 – typo 'HVSR'

L483 - this is an excellent finding. I think even more important given the findings of bedrock depth controls in this study.

L545 – this echoes my prior comment about 3D Valley composition.

L559 - I think the authors have enough data within this study to conceptualize a 'why'. Why does Staunton not dewater? I compiled some simple plots to illustrate potential interactions of interest. For instance, Staunton has both the deepest median DTB, but also the most confined valley (see Figures 1 to 3).

An additional plot of dtb, valley width, and volume of deposit in valley (dtb*width) reveals a negative correlation of temperature with dtb, a positive correlation of temperature with width and a strong negative correlation with volume. As such, it would seem the authors have stumble upon some zone of width/dtb ratio that offsets dewatering? I encourage the authors to dive a bit 'deeper' here, as I think they may have something novel to report here.

L574 – this is awesome!

L585 – okay, this is what I was speaking to earlier.

L597 - also, Paine run has median valley width 5 m > Staunton.

Figure 11 - might say GW influence 'prediction' in the legend too.

[Figure]

**DTB = *f*(valley width)**

**Figure 1** plotted relationship between depth to bedrock (DTB) and valley width for the Briggs et al. study. Where a strong negative correlation is found between DTB and valley width. I understand valley width is taken ~ 2 m above the valley floor, but this is still a meaningful measure.

[Figure]

**Figure 2** An overview of parameters for each study stream as per Table 1 in the MS

[Figure]

**Figure 3** I encourage the authors to think about the relationship between volume (storage) and the incised natured of the coldest streams, such as Staunton. Given these rich datasets the authors have generated, I think there is bandwidth to conceptualize what may be at play here. This will also allow the testing of these conceptual models in different settings.

---

## Author Comment (AC1)

**Please find our responses below in black font to each review point in red.**

**Review #1**
Briggs *et al.* (HESS) review
First, thank you for the opportunity to review this paper. I really enjoyed reading it. In their paper, Briggs *et al.* collated geophysical surveys, remote sensing data, and stream temperature and discharge loggers to reveal the role of bedrock depth, in catchments underlain by low hydraulic conductance bedrock, on stream dewatering and thermal resilience. The authors have done an excellent job in highlighting the role of fine-scale hydrogeological setting on the aforementioned hydrological processes. Further, the authors revealed that global scale datasets of depth to bedrock (DTB) largely overestimate this critical parameter, by as much as 12 m. Ultimately, this piece is very timely, and adds a nice story to the hydrology puzzle. I especially applaud the authors in identifying some complex processes that really drive home the importance of surface water -groundwater interactions.
I am happy to recommend this paper for publication after what I consider minor to moderate revisions. This dataset is incredibly rich, and whilst I understand it is not possible to do everything, I do think there is some bandwidth for the authors to dig a bit deeper into what is at play in Staunton. I think given the density of these data, a few things may be conceptualized. For instance, wider valleys will have high solar loading, and the recharge will be spread over a larger area. What role might this have on thermal and discharge regimes?
Thanks again, and nice work! Antóin O'Sullivan

Hello Antóin,
We sincerely appreciate your thoughtful and insightful review that was clearly informed by your own substantial research on similar physical watershed topics. The time you took to develop new plots of our data (e.g., valley width vs bedrock depth) is especially appreciated. You are correct in that we did not dive into some of the large-scale stream valley structural controls in detail. That choice was made in part to help keep the reader's focus on the extensive depth to bedrock and stream dewatering data, and in part because Johnson et al (2017) (doi.org/10.1016/j.jhydrol.2020.124929) explored numerous physical valley attributes in conjunction to explain stream temperature/groundwater influence patterns in these same catchments. However, we realize our current understanding of these systems could benefit from some additional analysis regarding physical valley controls on the shallow suprabedrock aquifers across the study catchments, including utilizing available LiDAR data. However, this type of analysis will need to be addressed as part of future work. The Shenandoah watersheds are extensively forested so we do not expect direct solar warming of the land surface to be a major factor in summer. In a revision of this discussion paper we plan to more thoroughly explore the relations between bedrock depth and valley width, valley width variability, and slope, both at local (to HVSR measurements) and average valley scales. We also note that the Johnson et al (2017) conducted in these same catchments found and inverse relation between valley width and their metrics of groundwater influence on stream temperatures, which agrees with the findings of the current study.

Apologies for grammatical errors herein.
Specific comments:

L 134: True, however, could this also be a function of bedrock *K*? For bedrock with a relatively high *K* (karst for instance), a duality may exist where a portion of the water is driven laterally - as stated - whilst another portion may be recharging the bedrock aquifer. These mechanisms are also likely temporally dynamic. In the setting of this study, I agree that lateral flow with bedrock shallowing seems most likely given low *K*. However, in the introduction, it may be best to speak exclusively to the conceptual controls in general.

Great point, as we want this introduction to be broader in scope than the hydrogeological setting of Shenandoah NP. This sentence was revised to include '..... and bedrock permeability.'

L 162: Not sure if it is worthwhile stating consolidated sediments, e.g., clay, may dampen the signal too (see Haefner, R.J., Sheets, R.A. and Andrews, R.E., 2010. Evaluation of the horizontal-to-vertical spectral ratio (HVSR) seismic method to determine sediment thickness in the vicinity of the South Well Field, Franklin County, OH.). If one assumes macro-pores in the clay are limited, this may also lead to low storage capacity in areas of shallow depth to low hydraulic conductance layer?

The HVSR technique is sensitive to the shear wave velocity used and in heterogeneous unconsolidated sediments there may be uncertainty associated with the technique. In the study cited, the influence of clay on depth to bedrock measured with HVSR occurs in an environment where 5-meter thick horizontal clay layers are present. In the study, these clay layers represent heterogeneity in the vertical shear wave profile. The authors note that "It is likely that the largest errors in sediment thickness arise from variability of geology (shear-wave velocity) in the subsurface". It is true that in the presence of a relatively impermeable clay lying above bedrock, one could estimate a depth to bedrock that wouldn't represent the depth to no-flow layer. However, we located our HVSR measurements within alluvial sediments and, based on available boring logs, feel confident that clay lenses such as this are not typical. We changed the sentence around L162 to include this new text: "While insensitive to variations in unconsolidated sediment permeability (i.e. identifying relatively impermeable clay layers), the HVSR method is effective at identifying the depth to distinct unconsolidated sediment/bedrock interfaces (Yanamaka et al., 1994)."

L 187: this is an awesome study site overview. nice job!  Thank you
L232: maybe spell out the acronym here as it is the first time it appears in a caption. done
L244: nice study sites! Thank you
L253: this progression seems logical As this was a multiyear study, we were able to adapt the study design year to year as we learned more about the geologic system.
L270: typo "bedrock". This aligns to my prior comment on the assumption the signal to being changed by bedrock. Hardpan or clay may also do this? This does not distract from the study, just worth a nod to the potential limitations of passive stratigraphy mapping

The HVSR method assumes a single shear velocity is representative of the depth profile; significant heterogeneity in subsurface material can lead to uncertainty in the depth estimates. In general, the impedance contrast between sandy soil and clay is not typically sufficient to lead to a resonance peak (even in the case of 5m thick horizontal clay layers as in Haefner, Sheets, and Andrews (2010)), so we believe we are justified in interpreting the resonance peaks to the bedrock interface, particularly as lenses of fines have not been documented for the Blue Ridge Mountain watershed systems previously. Additionally, our empirical and direct (active seismic) measures of shear wave velocity have been quite consistent across the NP.

L 271: and here is the answer, awesome :) Could there be an *n* shown for number of these boreholes? Looks to be *n*=6 from Goodling et al. report?
Correct- 'six boreholes' was inserted into this line, thank you.
L 290: typo 'teams' rectified
L301: nice! Yes, and it was great exercise.
L329: are these gauges co-located with temperature sensors as displayed in Figure 5? The gages are located a bit further downstream,
Figure 4 – this is a serious dataset, folks. Nice study design! Thank you
L352: smallest seems like an odd word for describing DTB, maybe lowest? This is not a game-changer, just a style thing. Yes there does not seem to be the perfect adjective here... but 'largest' seems to work for larger bedrock depths so 'smallest' seems the appropriate match for that term. 'Lowest' might be potentially interpreted as 'deepest'.
Table 1 - a quick regression of valley width to median bedrock depth illustrates a power law relationship (R2 ~0.62). Given how poor the broad scale depth to bedrock maps were at predicting bedrock depth, it may be useful to ill
ustrate here that using high res LiDAR and valley morphology as controls on bedrock rock may provide a more realistic view of bedrock depth in these areas. See Figure 1 below. Something for appendix maybe, but I think you have shown in a remarkably clear way that we need better geophysical data. Awesome stuff.
Thank you for explicitly pointing out the relation between average bedrock depth and valley width; in a revision we would add such a plot to our appendix material and discuss that finding in the main body text

Figure 5 – nice Thank you
L386 – might rephrase this sentence for clarity. done
Figure 7 - this is a powerful figure. It brings a lot of questions to mind. I wonder how this would look if one added another 2 panels that plotted the same dewatering observations and sub DTB with valley width? The reason I suggest this, the valley is 3D, by accounting for this 3D space, and given the authors have this amazing data set, it may point towards a more robust understanding of x,y,z space on these hydro processes. See Figures 2 and 3 below.
In a revision we would explore local variation in valley width along these two focus HVSR study reaches. One complicating factor are tributary confluences, which can substantially increase valley width at the ~100 m down valley scale. Both of these focus study reaches include tributary confluences.
L 435 – nice Thank you
L482 – typo 'HVSR' rectified
L483 - this is an excellent finding. I think even more important given the findings of bedrock depth controls in this study. Yes, given the average bedrock depth across all study watersheds was 3.4 m or smaller, this offset from the global scale dataset is stunning. Reviewer #3 correctly points out that we might not expect the global dataset to perform well in mountain regions with few borehole controls, and the authors of that dataset state as much in their paper; regardless, such interpreted bedrock data are currently being used to populate large-scale predictive models. Our work shows that approach is likely to be problematic if modeling goals include mountain baseflow dynamics and stream dewatering predictions.
L545 – this echoes my prior comment about 3D Valley composition. Agreed.
L559 - I think the authors have enough data within this study to conceptualize a 'why'. Why does Staunton not dewater? I compiled some simple plots to illustrate potential interactions of

interest. For instance, Staunton has both the deepest median DTB, but also the most confined valley (see Figures 1 to 3).

An additional plot of dtb, valley width, and volume of deposit in valley (dtb*width) reveals a negative correlation of temperature with dtb, a positive correlation of temperature with width and a strong negative correlation with volume. As such, it would seem the authors have stumble upon some zone of width/dtb ratio that offsets dewatering? I encourage the authors to dive a bit 'deeper' here, as I think they may have something novel to report here. Please see our response to our L585 comment below. There are some tantalizing potential research directions indicated in the existing data regarding bedrock depth and valley width, but we believe the hillslope recharge/storage dynamics must also be evaluated for such a more in-depth analysis of how these various physical stream valley controls interact to generate baseflow. Hillslope bedrock depth transect measurements are currently planned for the 2022 field season for the three focus subwatersheds (Paine, Piney, Staunton). Please stay tuned!

L574 – this is awesome! Thank you.

L585 – okay, this is what I was speaking to earlier. We have added to this statement regarding baseflow supply in Staunton: This apparent conundrum indicates the importance of bedrock depth (suprabedrock aquifer thickness) in facilitating spatially persistent baseflow generation during dry times.

L597 - also, Paine run has median valley width 5 m > Staunton. This detail was added to the sentence in question

Figure 11 - might say GW influence 'prediction' in the legend too. Agreed.

Figure 1 plotted relationship between depth to bedrock (DTB) and valley width for the Briggs et al. study. Where a strong negative correlation is found between DTB and valley width. I understand valley width is taken ~ 2 m above the valley floor, but this is still a meaningful measure.

Thank you for taking the time to plot these data together and for highlighting the apparent negative relation between the physical variables. We used a valley width of 2 m above valley floor so the measure would be less sensitive to fine scale topographic variation and instead better identify the true valley walls.

Figure 3 I encourage the authors to think about the relationship between volume (storage) and the incised natured of the coldest streams, such as Staunton. Given these rich datasets the authors have generated, I think there is bandwidth to conceptualize what may be at play here. This point is appreciated, but based on the lidar data from these catchments, we have not found systematic patterns in stream incision as related to summer temperature. We made some early attempts to predict measured bedrock depth by streambank height (ie channel incision) and that did not work for our test reaches, indicating stream incision is not directly related to bedrock depth in these coarse/rocky colluvial sediments. We might expect incision to be better related to bedrock depth, and therefore baseflow supply, in headwaters with fine valley sediments such as glacial till though fine sediments inherently have lower permeability and may inhibit groundwater exchange.

---

## Author Comment (AC2)

**Please find our responses below in black font to each review point in red.**

**Review #3**

Comment on hess-2021-622
Anonymous Referee #2
Referee comment on "Bedrock depth influences spatial patterns of summer baseflow, temperature, and flow disconnection for mountainous headwater streams" by Martin A. Briggs et al., Hydrol. Earth Syst. Sci. Discuss.,
https://doi.org/10.5194/hess-2021-622-RC3, 2022
Review of HES-2021-622 https://doi.org/10.5194/hess-2021-622

I appreciated the opportunity to review this interesting paper by Briggs and coauthors entitled 'Bedrock depth influences spatial patterns of summer baseflow, temperature, and flow disconnection for mountainous headwater streams'. The work addresses important questions regarding the description of connectivity and interaction between groundwater and surface water in mountainous catchments. The authors develop in their paper an interesting vision at the interfaces between geomorphology, hydrology and hydroecology (principally fish habitats). They performed systematic measurements of depth to bedrock along stream corridors in eight headwater streams in Shenandoah National Park (Virginia USA) using passive seismic technics along with identification of wet/dry segments and measurement of river temperature.

They highlight 3 main important outcomes from these measurements:
that measured bedrock depths strongly deviate from the ones available in global-scale geologic and soil dataset. permeable streambed thickness is highly discontinuous along the stream channels. On zones with important depth to bedrock, the authors identified localized disconnection of stream flow channels during extended period of droughts.
mean stream temperature during summer is negatively correlated with depth to bedrock suggesting preferential connectivity with groundwater with implications for stream aquatic ecosystems and habitats.

This paper has been carefully prepared and is well written. The introduction presents the context, state of the art and main questions in a comprehensive manner. The results are interesting and their interpretation are well supported by a robust analysis. The discussion and conclusions will definitely trigger the attention of the readers of HESS. I have only raised few general points and made suggestions that could be helpful for the authors to develop the discussion and conceptualization of their results.

We greatly appreciate your time and thoughtfulness on this review. You have accurately summarized the three main high-level points we intended to make in the manuscript, and are gratified you believe these findings will be of strong interest to the HESS readership.

I have some concerns regarding the comparison between measured depth to bedrock and the one compiled in global databases. I agree with the authors that such databases might not be suitable to capture local properties of soil types or depth to bedrock along the river corridor. Nonetheless, there is a major difference in representative scales

between the geophysical measurements and the estimates that are compiled in those databases. The depth to bedrock database from Shangguan et al. (2017) provides data over a spatial resolution of 250m, while the data presented here integrate a few cubic meters around the instrument (is the measurement scale actually mentioned in the manuscript?). I believe that it is still interesting to mention but I would recommend the authors to minimize its importance in the manuscript and acknowledge the main differences and complementarities between both datasets.

The HVSR data essentially represent a point measurement of bedrock depth, which as the reviewer correctly points out, is perhaps not directly comparable to the 250m scale global bedrock depth layer. However, we do believe that in aggregate the 191 HVSR measurements here are appropriate to compare to the corresponding 250m gridded data to assess general agreement in the data types. We have also shown in Figure A4 how bedrock depth is systematically overestimated by the global scale data layer. The point vs grid scale offset is difficult to escape, as the Shangguan bedrock depth model was created via using soil boreholes and well drilling observations, which also only integrate a limited spatial extent and could also be considered point measurements of bedrock depth.

In section 5.4 of the Shangguan and others (2017) paper, they note that the DTB predictions should be used with caution, particularly for shallow depths to bedrock and in mountainous regions. This is due to a reduction in variance (under prediction of deep DTB and overprediction of shallow DTB) resulting from the ML regression and due to a sparse number of observations in mountain ranges. In the discussion section, we now incorporate the Shangguan and others self-recognized limitation of their bedrock depth layer mountain settings such as Shenandoah NP>

We believe our study points to the additional work needed to characterize bedrock depth for headwater streams, and our comparison with the Shangguan dataset simply highlights problems that are likely for any global/regional model of bedrock depth based on limited borehole data. This 'point' is timely, as such large-scale geologic datasets are currently being used as templates for a range of predictive models that include baseflow generation dynamics.

It also remains unclear to me to what geomorphological processes/features of the landscape the measured depth to bedrock are assigned to: preferential erosion, fracturation/weathering, sediment accumulation, all of them without distinction? I believe that it would be important to link the measured stream corridor depth to bedrock and streamflow behaviors to some knowledge of local catchment-scale geomorphology/geology. This could help identifying generic information to be transferred to other catchments (or at least provide guidance). For example, in table 1, it seems that there is an inverse correlation between valley width and DTB. Also, one would expect that DTB impacts drainage density (dd~K*DTB) and intermittency (through aquifer volume available V~ 2*DTB*river length*hillslope length ~ DTB*river length/dd).

Exploring such generic relationship would help to conceptualize the results and increase the impact of the paper in my opinion.

The Reviewer is correct in that we generally did not attempt to identify the geologic mechanisms that controlled bedrock depth variability along the study streams or between them, except for Paine Run where there were more clearly pockets of alluvium and colluvium built up along a shallow, oft exposed, bedrock surface. We also speculate that the 20m+ deep apparent trough in the bedrock surface found along Piney River is an unmapped fault zone. We did not originally highlight the negative relation between measured bedrock depth and valley width but do so in the revised text. Assuming that the unconsolidated material along the valley floor is generally sourced from eroding hillslope colluvium, it does stand to reason that more narrow valleys show thicker deposits. While we understand that the omission of a more in depth analysis of bedrock depth controls for the study site may be unsatisfying to readers with geologic and geomorphic background and that the development of a transferrable relation could be useful to the community. In a revision of this paper we would explore rank order correlations between the physical valley variables already mentioned (bedrock depth, width means and variance; channel slope means and variance).

Some references:
Litwin et al 2021 https://doi.org/10.1029/2021JF006239 Great paper, we will add the citation to a revised version of the manuscript and take the findings into account when discussing our study.
Luo et al. 2010 https://doi.org/10.1130/G30816.1
Warix et al., 2021 https://doi.org/10.1002/hyp.14185 Great paper, we will add the citation to a revised version of the manuscript and take the findings into account when discussing our study.
Ilja van Meerveld et al. 2019 https://doi.org/10.5194/hess-23-4825-2019 We intended to cite this related paper in the original submission but that was somehow dropped along the process. We will add this paper to a revised version of the manuscript.

I believe that this work brings very interesting insights and data for our understanding of the impact of depth to bedrock to flow continuity and groundwater-surface water exchanges in mountain regions. I recommend the paper to be published in HESS.
Thank you.

Please
also consider few minor points listed in the following.
Specific comments:
l145-149: likely to be biased by the location of wells preferably implemented downhill and where more productive aquifer maybe be identified.
Agreed, this sentence was added: 'In more typical headwater systems, existing wells may be preferentially installed to maximize the production of water and not broadly sample the true range of bedrock depths.'
l167-169: do you mean in context where the water table is close to the surface? i.e. when

K/R (R=recharge) is low? The text 'for mountain stream corridors' was added to this statement for context, as when there is a stream present through permeable mountain sediments the water table is inherently close to/at the land surface (at the stream).

l234: how to differentiate sediment accumulation from weathering/fracturing development that can also enhance K? Sediment accumulation on a bedrock surface is expected to result in a clear/interpretable HVSR measurement, while heaving fracturing/weathering of the bedrock surface is expected to result in HVSR measurements that are of low confidence or are unable to be interpreted. That is because the method depends on distinct vertical changes in acoustic impedance. Therefore, our HVSR dataset is weighted toward evaluating zones of sediment accumulation on a low permeability bedrock surface as described generally in this paragraph.

l320: I did not understand how atmospheric effects were filtered here.

l326: providing the equation of BFI would help the readers that are not familiar with this Index BFI is not determined by a single equation, but through by connecting points along the hydrograph recession that are not expected to be impacted by storm/quickflow with straight lines and then calculating the daily ratio of baseflow to stormflow as distinguished by the lines. A reference to Barlow et al 2014 was added to this methods statement to aid the interested reader.

Figure 4: why showing depth in log here? I think it masks the actual variability of your Dataset. That is true regarding the deeper bedrock depth anomalies, but we found the log scale important in showing variation among the shallow bedrock depths that dominate the dataset.

Figure 4: 1 m seems to be the minimum depth measurable, correct? Is it mentioned in the manuscript? You may be correct, but we have not rigorously evaluated < 1m sediment thicknesses with the method and are not aware of other published results that address this. You do point to a potential issue, in that some number of measurements that we were not able to interpret may result from < 1m true bedrock depths, and those points would be preferentially dropped from the analysis.

table 1: it seems that there is an inverse correlation between valley width and DTB. Do you see correlation between drainage density and DTB? Since dd ~ K*DTB. It would be interesting to assess the relationship between landscape topography and measured DTB to identify generic relationship that could be transferred to other catchments. Absolutely, as noted above we plan to more explicitly describe and discuss the negative relation between valley width and bedrock depth, thank you for pointing that out. In this response we are assuming the reviewer is referring to "drainage density" (length of stream/drainage area) in the same way USGS sometimes uses drainage density as a basin characteristic when developing peak flow or low flow statistics. The first example that comes to mind is Bent and Archfield, 2002 (https://pubs.usgs.gov/wri/wri024043/pdfs/report.pdf) where they found that drainage density was inversely related to the probability that a stream flows perennially in Massachusetts in a wider/unconfined valley types. The study catchments are confined valley settings there is a relatively small range in drainage density between study sites; we feel that any relationship between DTB and drainage density that emerged might be applicable to only to small percentage of the otherwise wide range of drainage densities

possible in headwater streams but we will explore this topic in a revision of the manuscript.

l398: how is this analyzed/filtered? Text added: 'Paired air and water annual temperature signals exhibited a spectrum of shallow groundwater influences as indicated by extracting fundamental sinusoids from each multiyear temperature dataset per methods described by Briggs et al. (2018). Observed phase shifts between stream and local air annual temperature signals ranged from approximately 5 to 30 d with a mean of 11 d.'

Figure 8: I did not fully understand how this graph is interpreted. A complete interpretation of Figure 8a does potentially necessitate some additional background provided by Briggs et al 2018 and better cited in the revised manuscript.

l423: I did not fully understand what this means? Did you remove an outlier to improve Statistics? No, what we mean to indicate is the significance of the relationship is driven by the Staunton River data point, such that if that site was removed a linear relationship determined for the remaining seven sites is not significant.

Figure 9: it would be useful to add the confidence interval on this plot. We are not clear if the Reviewer suggests a confidence interval be added to the regression or the HVSR (or temperature) data points.

l450: I do not understand why "(low permeability)" is added between parenthesis here. Please clarify your meaning. These parentheses were removed. Our intention is to indicate that in settings of high bedrock permeability, bedrock depth may be a less-important control on shallow groundwater flowpath dynamics.

l479: they concern different spatial scales. Not sure how we can interpret this result. Please see our response to a similar point made above

general comments.

l495: I fully agree with this statement. However, the resolution of this database is way lower than the scale you are interested in. In consequence, it may appear obvious that differences exist. Yes, the disparity is perhaps expected/obvious, but in reality (in our experience) these types of large-scale geologic datasets are currently being used to inform models of subbasin scale GW/SW exchange processes. As our study is unique in measuring bedrock depth at relatively high spatial resolution along several mountain streams we feel these comparisons are worthwhile. We do not mean to negate the general value of the large-scale datasets but instead point to important challenges in specifying the geology of mountain stream networks.

l619: I find hazardous to compare two different years with different recharge records. The BFI is integrative of full baseflow period, but may not be representative of the punctual measurement performed. Could you clarify this point? We calculate BFI on seasonal timescales for this study and report the values from 2015 and 2019 here to show the similarity in calculated baseflow fraction such that these two years were comparable from a baseflow perspective for the purposes of discussion.